# HMGB1 signaling phosphorylates Ku70 and impairs DNA damage repair in Alzheimer's disease pathology

Hikari Tanaka[1,10], Kanoh Kondo[1,10], Kyota Fujita[1,10], Hidenori Homma [1,10], Kazuhiko Tagawa[1], Xiaocen Jin[1], Meihua Jin[1], Yuki Yoshioka[1], Sumire Takayama[1], Hitomi Masuda[2], Rie Tokuyama[2], Yukoh Nakazaki[2], Takashi Saito[3], Takaomi Saido [4], Shigeo Murayama[5,6], Teikichi Ikura [7], Nobutoshi Ito[7], Yu Yamamori[8], Kentaro Tomii[8], Marco E. Bianchi [9] & Hitoshi Okazawa [1✉]

DNA damage is increased in Alzheimer's disease (AD), while the underlying mechanisms are unknown. Here, we employ comprehensive phosphoproteome analysis, and identify abnormal phosphorylation of 70 kDa subunit of Ku antigen (Ku70) at Ser77/78, which prevents Ku70-DNA interaction, in human AD postmortem brains. The abnormal phosphorylation inhibits accumulation of Ku70 to the foci of DNA double strand break (DSB), impairs DNA damage repair and eventually causes transcriptional repression-induced atypical cell death (TRIAD). Cells under TRIAD necrosis reveal senescence phenotypes. Extracellular high mobility group box 1 (HMGB1) protein, which is released from necrotic or hyper-activated neurons in AD, binds to toll-like receptor 4 (TLR4) of neighboring neurons, and activates protein kinase C alpha (PKCα) that executes Ku70 phosphorylation at Ser77/78. Administration of human monoclonal anti-HMGB1 antibody to post-symptomatic AD model mice decreases neuronal DSBs, suppresses secondary TRIAD necrosis of neurons, prevents escalation of neurodegeneration, and ameliorates cognitive symptoms. TRIAD shares multiple features with senescence. These results discover the HMGB1-Ku70 axis that accounts for the increase of neuronal DNA damage and secondary enhancement of TRIAD, the cell death phenotype of senescence, in AD.

[1] Department of Neuropathology, Medical Research Institute and Center for Brain Integration Research, Tokyo Medical and Dental University, 1-5-45 Yushima, Bunkyo-ku, Tokyo 113-8510, Japan. [2] Chiome Bioscience, Inc., Sumitomo Fudosan Nishi-shinjuku Building No. 6, 3-12-1 Honmachi, Shibuya-ku, Tokyo 151-0071, Japan. [3] Department of Neurocognitive Science, Institute of Brain Science, Nagoya City University Graduate School of Medical Sciences, 1 Kawasumi, Mizuho-ku, Mizuho-cho, Nagoya, Aichi 467-8601, Japan. [4] Laboratory for Proteolytic Neuroscience, RIKEN Center for Brain Science, 2-1 Hirosawa, Wako, Saitama 351-0198, Japan. [5] Department of Neuropathology, Tokyo Metropolitan Institute of Gerontology, 35-2 Sakae-cho, Itabashi-ku, Tokyo 173-0015, Japan. [6] Brain Bank for Neurodevelopmental, Neurological and Psychiatric Disorders, Molecular Research Center for Children's Mental Development, United Graduate School of Child Development, Osaka University, Suita, Osaka, Japan. [7] Department of Structural Biology, Medical Research Institute, Tokyo Medical and Dental University, 1-5-45 Yushima, Bunkyo-ku, Tokyo 113-8510, Japan. [8] Intelligent Bioinformatics Research Team, Artificial Intelligence Research Center, National Institute of Advanced Industrial Science and Technology, 2-4-7, Aomi, Koto-ku, Tokyo 135-0064, Japan. [9] Division of Genetics and Cell Biology, IRCCS San Raffaele Scientific Institute, Milan, Italy. [10]These authors contributed equally: Hikari Tanaka, Kanoh Kondo, Kyota Fujita, Hidenori Homma. ✉email: okazawa-tky@umin.ac.jp

Impairment of DNA damage repair leads to accumulation of DNA damage in neurons of multiple neurodegenerative diseases such as Huntington's disease (HD) and spinocerebellar ataxias (SCAs)[1–12]. Moreover, human genome researches revealed DNA repair genes as modifiers of the progression of HD and SCA1[13–15]. The accumulating evidences indicate impairment of DNA damage repair as a common pathological domain across neurodegenerative diseases[15,16]. DNA damage is also increased in Alzheimer's disease (AD)[17,18], and hyperactivity of neurons in the visual cortex of AD model mice by photic stimulation is correlated to the extent of DNA damage[19,20], possibly reflecting the vulnerability of hyperactive brain regions such as default mode network in human AD. However, the mechanism underlying the increase of DNA damage in AD has not been unraveled.

DNA damage accumulates also in senescent cells, which secrete a set of chemokines and cytokines collectively known as senescence-associated secretory phenotype (SASP)[21–24]. SASP is recently implicated in stress-induced senescence[25–27] in addition to its oncogenic roles[28,29]. Moreover, the presence of senescent glial and neuronal cells has been reported in neurodegeneration and the aging brain[30–35]. SASP includes the chromatin protein HMGB1[36], which is also a damage-associated molecular pattern (DAMP) or alarmin. HMGB1 alerts the local cellular environment and distant cells of the immune system to the death or ongoing critical stress of individual cells. Furthermore, HMGB1 is passively released from necrotic neurons to extracellular space at the very early AD stage[37], and repetitive depolarization induces live neurons to actively release HMGB1[38]; HMGB1 is also involved in the pathogenesis of epilepsia[39]. For all these reasons, we investigated whether HMGB1 might be involved as an active factor in the DNA damage associated to neurodegeneration.

These lines of results prompted us to hypothesize that DNA damage might spread by a diffusible factor. Here we report that extracellular HMGB1 engages the receptor Toll-like receptor 4 (TLR4) and its downstream signaling to phosphorylate specifically serines 77 and 78 of the DNA repair protein Ku70, which is required for non-homologous end-joining (NHEJ), the major pathway of double-strand break (DSB) repair in neurons. HMGB1-induced phosphorylation of Ku70 impairs its binding to DNA and assembly into DSB foci and increases the number of DSB by reducing their repair. In vivo administration of anti-HMGB1 antibody (HMGB1-Ab) to postsymptomatic AD model mice decreases the number of neuronal DSB and ameliorates symptomatic progression. These results indicate that extracellular HMGB1 is the critical mediator causing the DNA damage propagation among neurons in AD pathology, via phosphorylation of Ku70.

## Results

**Impairment of DNA damage repair by HMGB1-induced phosphorylation of Ku70 at Ser77/78 under AD pathology**. Comprehensive and quantitative phosphoproteome analysis of brain tissues of human AD postmortem patients reveals some specific proteins were phosphorylated from the early phase of AD[40]. We noticed abnormal phosphorylation of Ku70 at Ser77/78 in human postmortem AD brains (Fig. 1a), but had not extended investigation. This time, we performed comprehensive and quantitative phosphoproteome analysis of human U2OS cells in the presence of HMGB1 in culture medium. Unexpectedly, extracellular HMGB1 induced the similar phosphorylation of Ku70 at Ser77/78 (Fig. 1a). Moreover, the inhibiting antibody (Fig. 1a), human anti-human disulfide HMGB1-Ab (dsHMGB1-Ab), which efficiently prevented HMGB1 from interaction with TLR4, inhibited the phosphorylation of Ku70 at Ser77/78

(Fig. 1a). These findings prompted us to further investigate the role of Ku70 phosphorylation at Ser77/78 in the AD pathology.

A previous study determined the crystal structure of Ku70 binding to double-strand DNA[41] and revealed that Ser78 is located at the surface of Ku70 protein directly contacting with phosphates on DNA[41]. Consistently, simulation using the PyMOL software supported that phosphorylation at Ser77 and Ser78, which are conserved between human and mouse (Supplementary Fig. 1), affects binding force between Ku70 and DNA by increasing negative charges of Ser77/78 and by disrupting the hydrogen bond (Fig. 1b and Supplementary Fig. 2).

Thr90, whose phosphorylation was increased in human AD brains and induced by extracellular HMGB1 (Fig. 1a), was far from the DNA interaction surface of Ku70 (Supplementary Fig. 2). Thr401 and Ser520, whose phosphorylation was increased in human AD brains but not stimulated by HMGB1 in U2OS cells (Fig. 1a), were not good candidates to affect Ku70–DNA interaction. The distance between Thr401 and DNA was 7.8 Å, which is too remote to generate a hydrogen bond (Supplementary Fig. 2).

Molecular dynamics (MD) simulation by super computers at The University of Tokyo (Shirokane) and National Institute of Advanced Industrial Science and Technology (ABCI 2.0) predicted the change of distance between Ku70 and DNA, further supporting the hypothetical downregulation of Ku70 access to DNA by phosphorylation of Ser77/78 (Fig. 1c). Consistently, gel mobility shift assay with phospho-mimetic mutants of Ku70 revealed that phosphorylation at Ser77 and Ser78 decreased the affinity to double-strand DNA (Fig. 1d).

**Ku70 phosphorylation at Ser77/78 prevents deacetylation by SIRT1**. In addition to the decreased affinity of pSer77/78 Ku70 for DNA (Fig. 1c), we found that pSer77/78- and pSer77-mimetic mutants of Ku70 were significantly delayed in localizing to DSBs generated by laser microirradiation, remaining in speckles around the nucleus (Fig. 2a). Such a delay in mobilizing Ku70 prompted us to hypothesize that the phosphorylation of Ser77/78 may affect protein–protein interactions (PPIs) of Ku70, which may otherwise aid localization to DNA damage foci. Previous reports suggest VCP[42], TDP43[43], 14-3-3[44], and SIRT1[45–47], an NAD-dependent deacetylase reported to deacetylate Ku70[45,46], as modulators of Ku70. Immunoprecipitation of overexpressed wild-type or phospho-mimetic Ku70 mutants in ultraviolet (UV)-irradiated U2OS cells did not pull down endogenous VCP or TDP43, excluding them as candidate Ku70 modulators (Fig. 2b). Meanwhile, the physical interaction between Ku70 and both Ku80 and DNA PKcs, which together form a complex at broken DNA ends during DSB repair, was not influenced by Ku70 phosphorylation (Fig. 2b). However, pSer77/78 mimetic Ku70 mutants (EGFP-Ku70-SS77/78DD and EGFR-Ku70-SS77/78EE) co-immunoprecipitated 14-3-3 but not SIRT1, whereas a mutant that cannot be phosphorylated at Ser77 or Ser78 (EGFP-Ku70-SS77/78AA) co-immunoprecipitated SIRT1 alone (Fig. 2b). Interestingly, Ku70 pSer77/78 mimetic mutants were found to be acetylated at Lys331 (Fig. 2b), a residue that faces Ser77/78 intramolecularly across the grove in which DNA sits (Fig. 2c). These data suggest a model in which 14-3-3 binds Ku70 pSer77/78, preventing the SIRT1 interaction with Ku70 and the acetylation of Ku70 Lys331 (Fig. 2d). Residual acetylation at Lys331 could delay dynamics of Ku70 to DNA damage foci as proposed[45,46] and/or reduce affinity to DNA as reported previously[48].

**Characterization of human anti-human dsHMGB1-Ab**. To prepare an efficient tool to interrupt the HMGB1 signal, we generated an anti-human dsHMGB1-Ab. Employing the

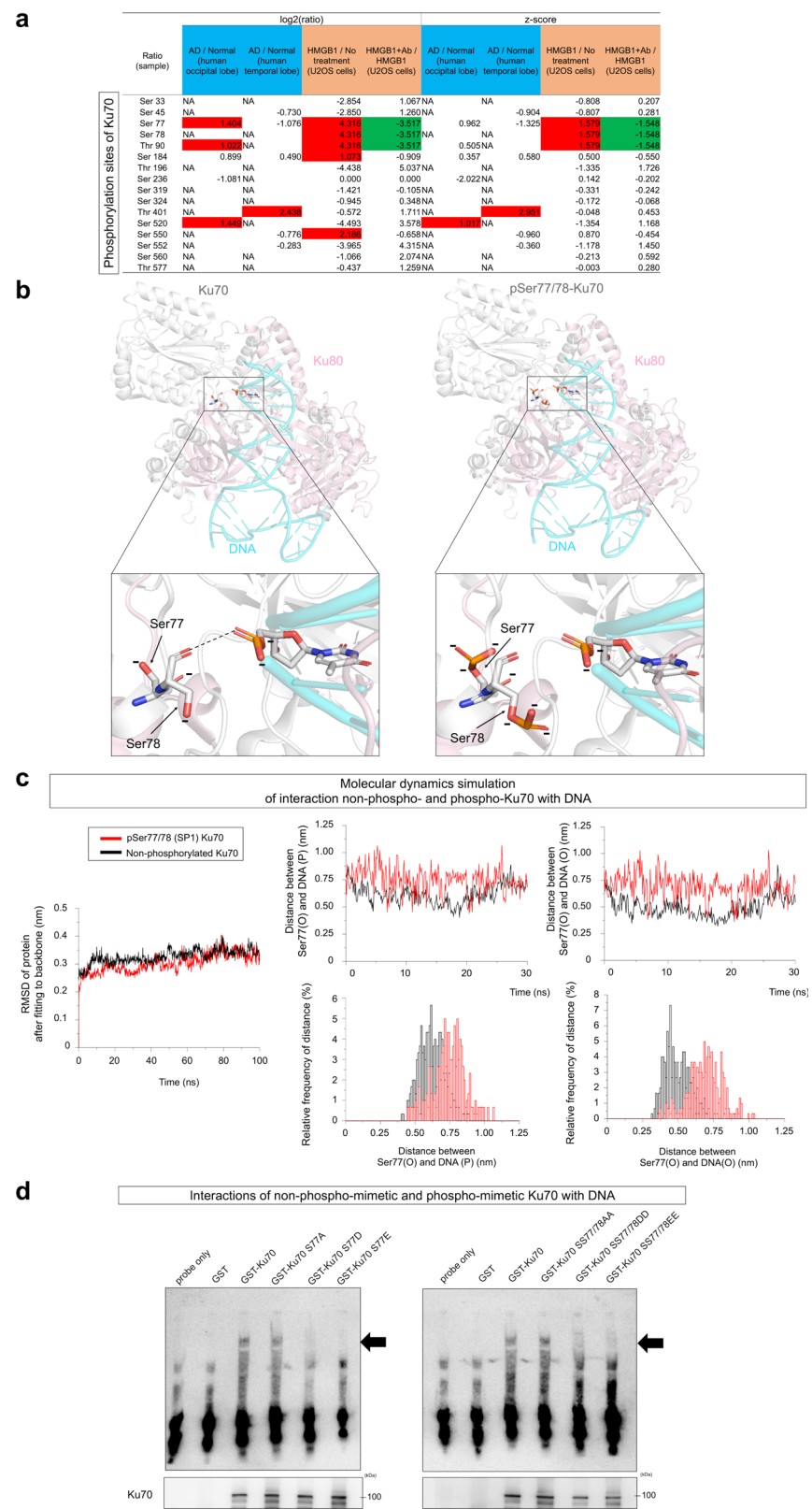

Autonomously Diversifying Library (ADLib®) system[49,50], we intended to obtain human monoclonal antibodies (mAbs) against human dsHMGB1, which efficiently interact with HMGB1, suppress HMGB1-induced MARCKS (myristoylated alanine-rich C kinase substrate)-Ser46 phosphorylation, and inhibit HMGB1-receptor interaction(s). More than $1 \times 10^7$ unique clones were screened via the ADLib® system by Chiome Bioscience, Inc., and

the positive clones were selected via first and second screenings of enzyme-linked immunosorbent assay (ELISA), western blotting with purified human dsHMGB1 (Supplementary Fig. 3a–c), and functional assay to examine their suppressive effect on HMGB1-induced MARCKS phosphorylation in primary cortical neurons (Supplementary Fig. 3d). The four selected mAbs were further evaluated via surface plasmon resonance (SPR) (Supplementary

**Fig. 1 Ku70 phosphorylation at Ser77/78 impairs binding to DNA damage foci. a** Increased phosphorylation of Ku70 at Ser77/78 or Thr90 was observed commonly in human AD brains (mean, $N = 5$) and HMGB1-treated human U2OS cells (mean, $N = 16$). Co-addition of anti-HMGB1 antibody suppressed the increase of phosphorylation at Ku70 at Ser77/78 or Thr90. Phosphorylation at Thr401 and Ser520 was increased in human AD brains but not stimulated by HMGB1 in U2OS cells. Phosphorylation at Ser550 was stimulated by HMGB1 in U2OS cells but not increased in human AD brains. **b** Based on Ku70 structure at PDB (ID: 1JEY) (left), structure of Ku70 phosphorylated at Ser77 and Ser78 was modeled by PyMOL (Shrödinger, LLC) (right), suggesting that phosphorylation at Ser77 and Ser78 increases their polarity and interrupts hydrogen bond at the interaction surface between Ku70 and DNA. The positions of Thr90, Thr401, and Ser520 in the Ku70–DNA complex model are close to Ku70–DNA interaction surface as indicated in Supplementary Fig. 2. **c** Molecular dynamics (MD) simulation. Left panel shows root mean square difference (RMSD) reflecting the squared average of 3D positional changes (coordinate values) of all atoms in the Ku70–Ku80 protein complex. The graphs of non-phosphorylated Ku70 and pSer77/78 Ku70 (SP1) revealed stabilities of their structures. Middle panels show the distance (nm) and relative frequency of distance between Ser77(O) and DNA-thymidine4(P) during 30 ns, and right panels show those of Ser77(O) and DNA-thymidine4(O1P). MD simulation suggested that the distance of pSer77/78 Ku70 from DNA was larger than that of non-phosphorylated Ku70. **d** Gel shift assay was performed with GST-fusion proteins of phospho-mimetic and non-phospho-mimetic mutants of Ku70 at Ser77/78.

Fig. 3e) and antibody #129 was identified as the best candidate for further evaluation, because it suppressed HMGB1-induced phosphorylation of MARCKS at Ser46 in primary mouse cortical neurons most efficiently (Supplementary Fig. 3d) and because its $K_D$ was the lowest among the selected antibodies (Supplementary Fig. 3e). Antibody #129 interacted with human dsHMGB1 at a high affinity but not with human all-thiol-HMGB1 (Supplementary Fig. 3e), which interacted similarly with mouse dsHMGB1 (Supplementary Fig. 3e). Moreover, the SPR analysis confirmed that antibody #129 inhibited the interaction between TLR4 and human dsHMGB1 (Supplementary Fig. 3f). At a 6 : 1 molar concentration ratio of dsHMGB1 to human anti-HMGB1-Ab, the dsHMGB1-TLR4 interaction was suppressed to 30% of that in the absence of the antibody (Supplementary Fig. 3f, g). Furthermore, antibody #129 also inhibited the interaction between receptor for advanced glycation endproducts (RAGE) and dsHMGB1 (Supplementary Fig. 3g, h). The $K_i$ of antibody #129 for TLR4 calculated from the SPR data was 3.32 nM (Supplementary Fig. 3h) and lower than the $K_i$ for RAGE (7.78 nM) (Supplementary Fig. 3h).

To re-confirm the $K_D$ value obtained from SPR, we performed ELISA-based calculation of $K_D$ value by Michaelis–Menten equation (Supplementary Fig. 3i). The $K_D$ value obtained from ELISA ($5.78 \times 10^{-11}$ M) matched quite well with the value obtained from SPR ($1.47 \pm 1.16 \times 10^{-11}$ M), considering the indirect fixation of human dsHMGB1 to the bottom of the ELISA well via anti-HMGB1-Ab in comparison to the direct fixation of human dsHMGB1 to the SPR chip.

ELISA-based epitope mapping with synthetic peptides revealed that the antigen region (AFFLFCSEYR) of antibody #129 overlapped with a TLR4-binding site that has been reported in other studies[51–53] (Supplementary Fig. 3j). Furthermore, the antibody bound weakly to two other regions (YAFFVQTCRE and VNFSEFSKKC). When mapped to a HMGB1 three-dimensional structure (Protein Data Bank (PDB) ID: 2YRQ), the two antigen regions (YAFFVQTCRE and AFFLFCSEYR) were located on the first helices of the HMGB1 Box A and Box B domains, respectively (Supplementary Fig. 3k, l), suggesting that antibody #129 binds to the regions via homologous interactions. Another weak antigenic region (VNFSEFSKKC) was found in the second helix of Box A (VNFSEFSKKC), which is in direct contact with YAFFVQTCRE. It is expected that antibody binding to these regions will alter the HMGB1 structure and decrease its affinity to TLR4. These data consistently indicated that antibody #129 might efficiently suppress the signaling pathway triggered by dsHMGB1-TLR4 binding.

**BBB transfer of human monoclonal anti-human HMGB1-Ab.** The antibody #129 was revealed to transfer across the blood–brain barrier (BBB) based on multiple experimental results

(Supplementary Fig. 4). In both cases of subcutaneous and intravenous administration (Supplementary Fig. 4a), immunohistochemistry with 3,3′-Diaminobenzidine (DAB) and fluorescence revealed transfer of the antibody #129 into the brain parenchyme of normal mice (Supplementary Fig. 4b, c). The transfer into parenchyme including neuropil and cell body seemed higher in 5xFAD (Familiar Alzheimer Disease) than normal background mice (Supplementary Fig. 4d). We further confirmed the transfer across BBB by conjugating biotin to antibody #129 before subcutaneous and intravenous injections (Supplementary Fig. 4e), which enabled us to detect the antibody specifically and sensitively in the brain parenchyme by immunohistochemistry (Supplementary Fig. 4f) and ELISA (Supplementary Fig. 4g). In this experimental system, the ratio of antibody concentration in the cortex per plasma was about 2–8%, whereas the ratio of total amount of antibody in the cortex per plasma was 0.3–0.7% (Supplementary Fig. 4h).

**Interruption of HMGB1 signal recovers neuronal DSB.** Immunohistochemistry of γH2AX and 53BP1, markers for DSB, revealed increased DSB accumulation in the cerebral cortex of 5xFAD mice, whereas subcutaneous administration of human monoclonal anti-HMGB1-Ab #129 clearly suppressed DSB accumulation (Fig. 3a). This finding was further confirmed by western blot analyses (Fig. 3b). Similarly, intravenous administration of #129 antibody suppressed DNA damage in cortical neurons of 5xFAD mice (Fig. 3c, d). Consistently, DNA damage was increased in MAP2-positive neurons at the parietal cortex of human AD patients (Fig. 3e).

As Ku70 senses DSB in the NHEJ pathway, we further tested how extracellular HMGB1-triggered signals affect Ku70 dynamics in human U2OS cells endogenously expressing the TLR4 (https://www.proteinatlas.org/ENSG00000136869-TLR4/cell) and RAGE receptors (https://www.proteinatlas.org/ENSG00000204305-AGER/cell). We performed time-lapse imaging of EGFP-Ku70-expressing cells after laser microirradiation. To select the HMGB1 concentration for the U2OS cell cultures, we evaluated the HMGB1 concentration in the cerebrospinal fluid (CSF) of human AD patients ($n = 56$) via ELISA and determined that the CSF HMGB1 concentrations ranged from 11 pg/mL to 13.7 ng/mL, mean = 936 pg/mL[37]. Therefore, we used 6 ng/mL in this in vitro experiment as a model for the HMGB1 concentration in the neuropil space following neuronal damage.

Under these conditions, HMGB1 addition to the medium significantly delayed the EGFP-Ku70 accumulation at damage sites (Fig. 3f, g). It is noteworthy that the Ku70 reservoir area was depleted after laser microirradiation (Fig. 3f, arrows), whereas HMGB1 addition inhibited the recruitment of Ku70 from the reservoir area (Fig. 3f). Co-addition of human monoclonal anti-HMGB1-Ab (#129) almost entirely rescued the

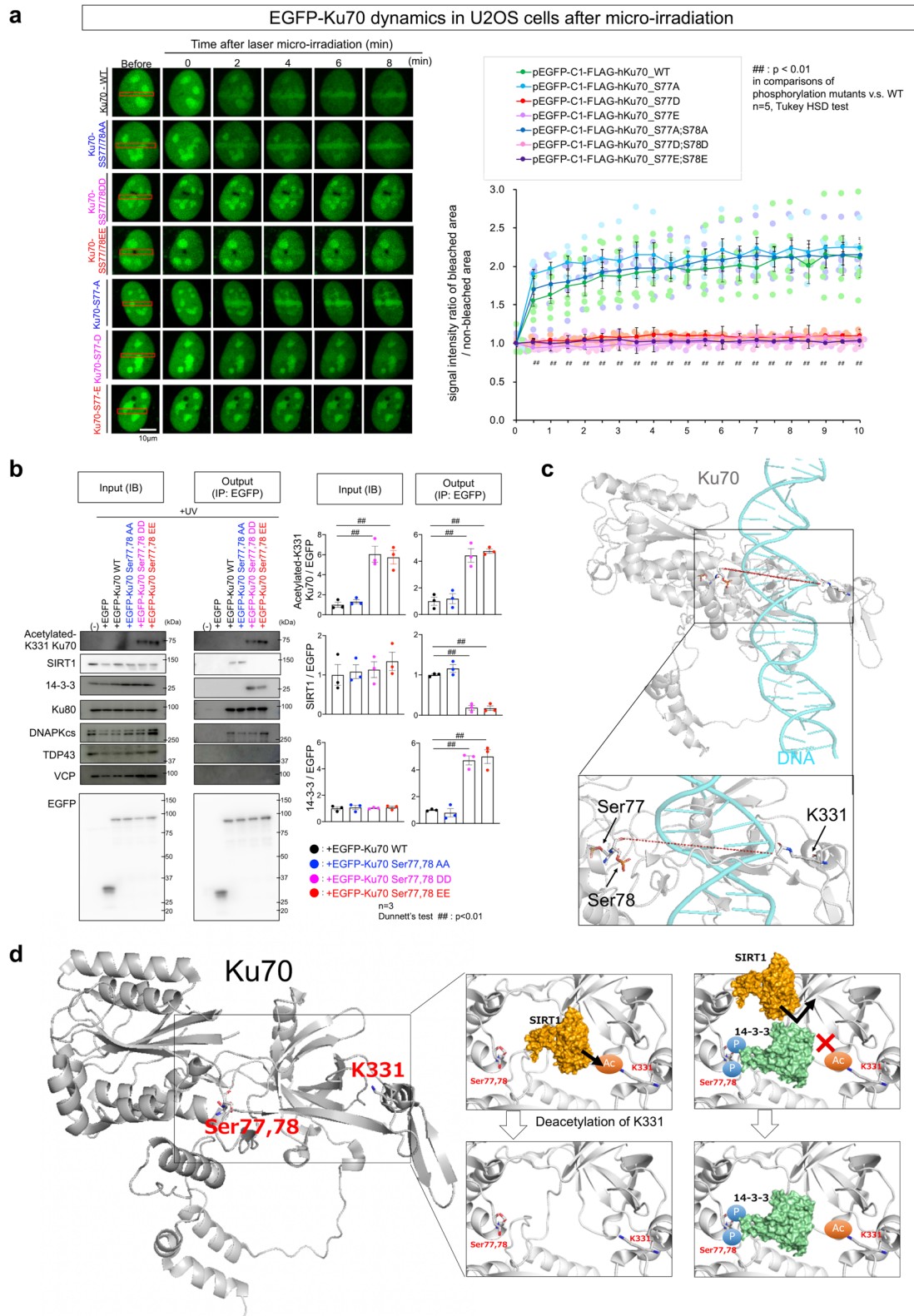

**Fig. 2 Ku70 phosphorylation at Ser77/78 impairs Ku70 foci formation after DNA damage and prevents deacetylation by SIRT1. a** Ku70 phosphorylation at Ser77/78 impairs dynamics to DNA damage foci. Upper panels show accumulation of wild-type, non-phospho-mimetic mutant, and phospho-mimetic mutants of Ser77/78 or Ser77 to DNA damage foci in U2OS cells after microirradiation. Values in each group are summarized by mean ± SEM. **b** Co-immunoprecipitation of phospho-mimetics of Ku70 reveals a physical interaction between Ku70 and either SIRT1 or 14-3-3. Phosphorylation of Ku70 results in a change of acetylation state. The phosphorylation state of Ku70 does not affect interactions with Ku80 or DNA PKcs. Ku70 does not interact with TDP43 or VCP. Right graphs show quantitative analyses of three independent experiments ($n = 3$, ##$p < 0.01$ in Dunnett's test). **c** Positional relationship between Ser77/78 and Lys331 in the Ku70 structure when bound to DNA (PDB: 1JEY). **d** Hypothetical model depicting the competitive interactions of SIRT1 and 14-3-3 with Ku70 and the resultant acetylation of Lys331.

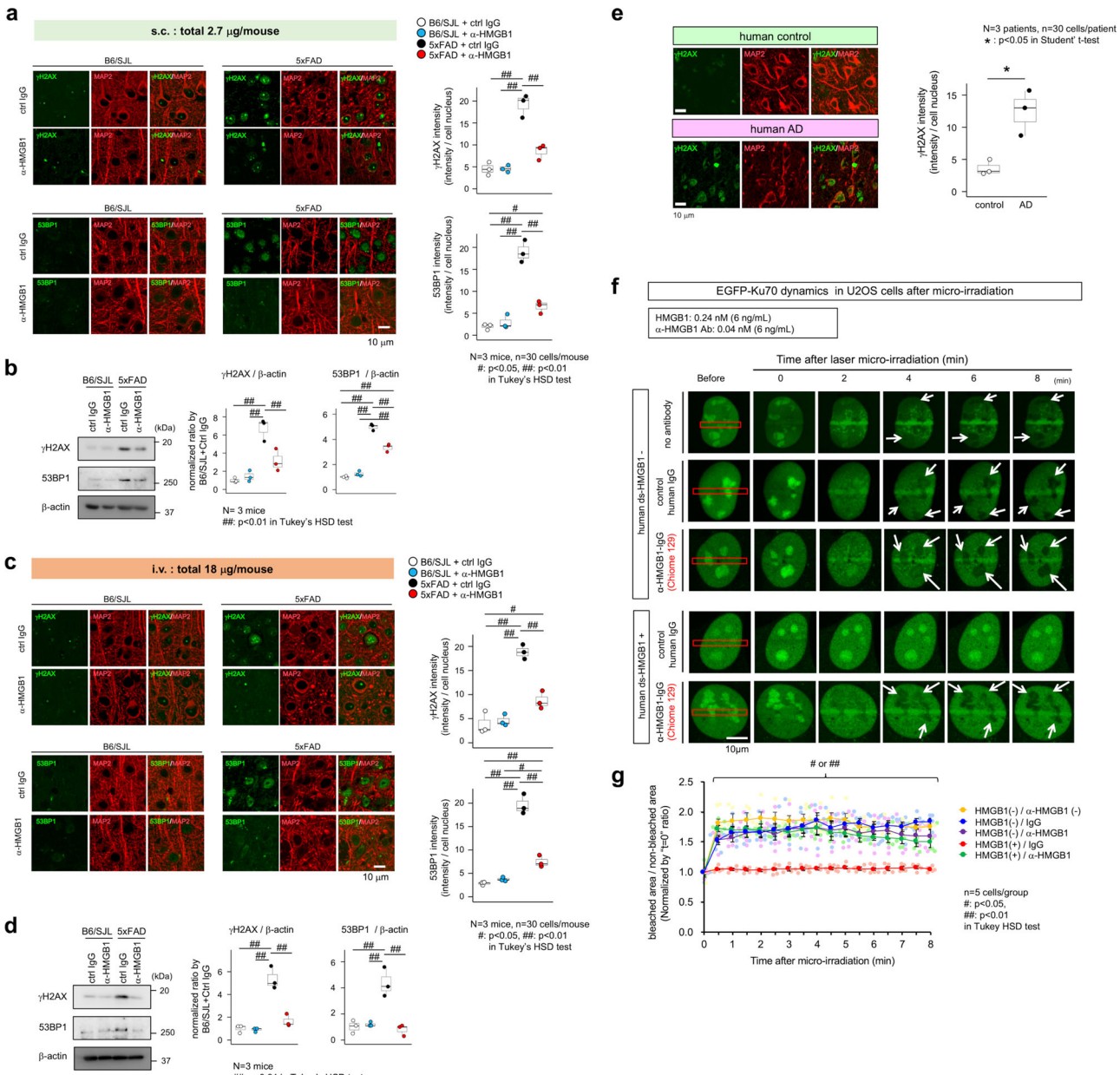

**Fig. 3 Interruption of HMGB1 signal recovers neuronal DSB. a** The abundance of DNA double-strand breaks was evaluated using two markers (γH2AX and 53BP1) in HMGB1 antibody-treated or control IgG-treated 5xFAD mice and non-transgenic sibling mice (B6/SJL). The graphs on the right show quantifications of the intensity/nuclei for the four groups. $N = 3$ mice, $n = 90$ cells. Tukey's HSD test revealed increased DNA damage in 5xFAD mice and a rescue via treatment with human monoclonal anti-HMGB1 antibody. **b** Western blot analysis of the entire cerebral cortices from the four groups with anti-γH2AX and anti-53BP1 antibodies ($N = 3$). **c** Immunohistochemical staining of γH2AX and 53BP1 in occipital cortex neurons of 5xFAD mice that received subcutaneous or intravenous administration of human monoclonal anti-HMGB1 antibody (#129) or control human IgG. The graphs on the right show quantitative analyses of the γH2AX and 53BP1 signals in MAP2-positive neurons. **d** Western blot analyses of γH2AX and 53BP1 levels in whole cortex samples. **e** Immunohistochemical staining of human postmortem brain tissue samples (occipital lobe) from patients with no neurological diseases and AD patients with anti-γH2AX antibody. **f** The dynamics of Ku70 recruitment to DNA damage foci induced via microirradiation were evaluated by transiently expressing EGFP-Ku70 in U2OS cells cultured in the presence/absence of HMGB1 (0.24 nM). Human monoclonal anti-HMGB1 antibody (#129) but not control human IgG or mock treatment rescued the delayed EGFP-Ku70 accumulation at DNA damage foci. **g** Quantitative analyses of the EGFP-Ku70 intensities at damage sites from time-lapse images of U2OS cells ($n = 5$) from which the EGFP-Ku70 intensities from a non-damaged area of the nucleoplasm were subtracted. The rate of EGFP-Ku70 accumulation was reduced in the presence of HMGB1 but recovered upon co-addition of human anti-HMGB1 monoclonal antibody to the culture medium. Values in each group are summarized by mean ± SEM. Box plots show the median, quartiles, and whiskers that represent data outside 25th to 75th percentile range.

impaired EGFP-Ku70 dynamics (Fig. 3f). Quantitative analyses of the signal intensities at the damage sites confirmed these results (Fig. 3g). Collectively, these data indicated that interruption of HMGB1 signal recovers neuronal DSB.

**PKC–MEK–ERK pathway mediates the HMGB1 signal to Ku70.** Next, we addressed upstream signals leading to Ku70 phosphorylation by using the data from comprehensive phosphoproteome analysis of the cerebral cortex tissue samples of

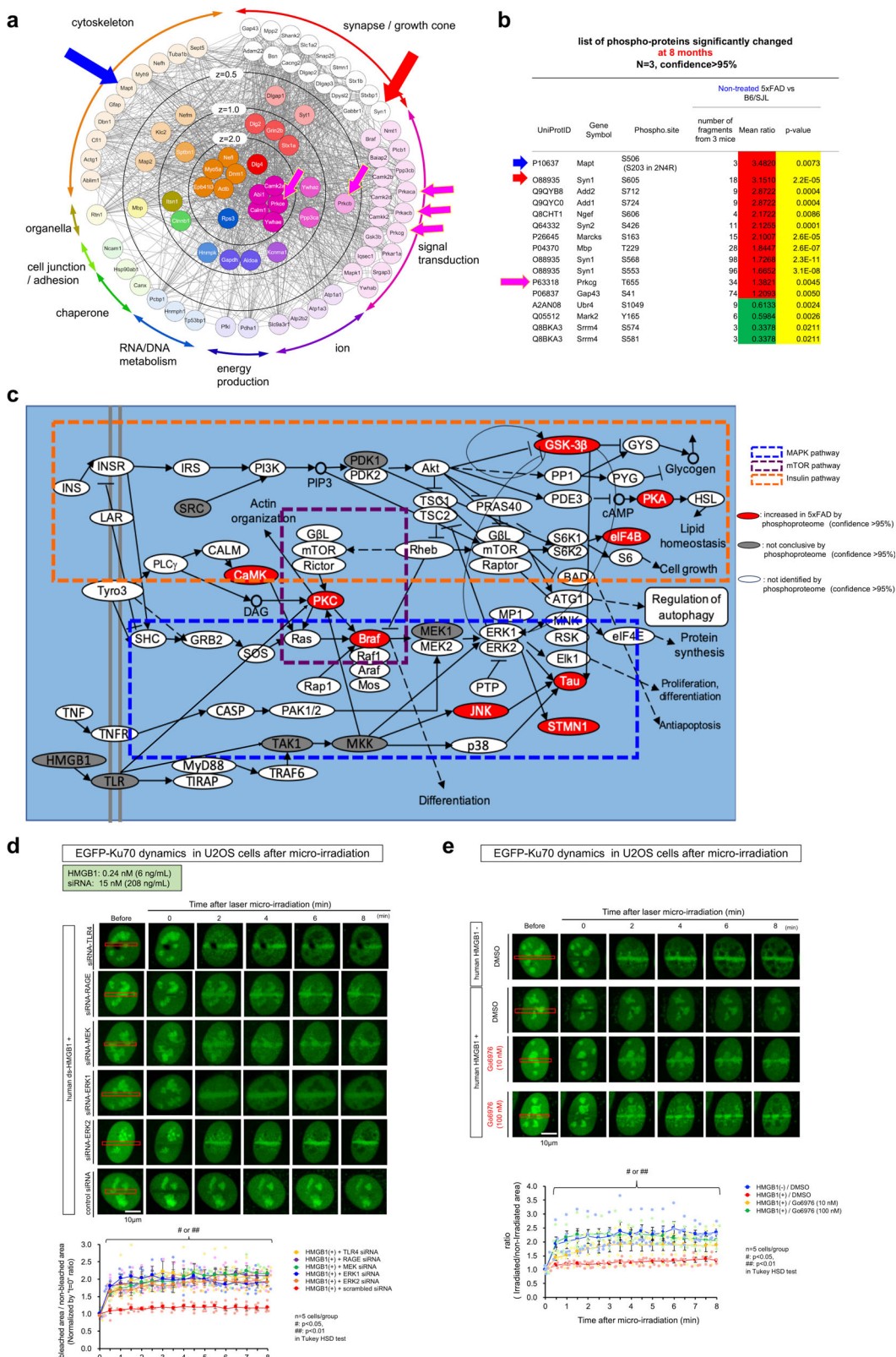

non-treated B6/SJL mice, non-treated 5xFAD mice, and HMGB1-Ab-treated 5xFAD mice (Fig. 4). Phosphorylated peptides and proteins detected with a reliability of >95% in our comprehensive phosphoproteome analysis (Supplementary Fig. 5a) were mapped onto PPI network in the case significantly changed in at least one molecular network of AD mice (Supplementary Fig. 5b), extracted the core signaling network based on the centrality of proteins

(betweenness scores), and classified the functional properties in the core signaling network (Fig. 4a). The result indicated $Ca^{2+}$/calmodulin-dependent protein kinase IIa, other Camk2 paralogs, protein kinase Cβ (PKCβ), PKCε, and various types of cAMP-dependent protein kinases (PKAs) as key molecules in the pathological network (Fig. 4b), and suggested either mitogen-activated protein kinase (MAPK) pathway composed of PKC,

**Fig. 4 MAPK pathway mediates the HMGB1 signal to Ku70. a** A core signal network generated based on the core proteins with high centrality scores selected from the phosphoprotein network in Supplementary Fig. 5b and repositioned according to their functions. The proteins were functionally categorized into the follow classes: signal transduction, synapse, cytoskeleton, organelle, cell junction, energy production, DNA/RNA metabolism, ion channel, and chaperone. **b** Phosphorylation of tau and synapse-related proteins became remarkable at the late stage of pathology (8 months of age) in 5xFAD mice. **c** Phosphorylation changes mapped onto KEGG database indicate activation of MAPK signaling pathway. **d** Upper panels show the influence of siRNAs against kinases in MAPK pathway on the Ku70 accumulation to DNA damage foci. The impaired accumulation of Ku70 in the presence of HMGB1 was rescued by siRNAs against MAPK pathway kinases. SignalSilence® Control siRNA (Cell Signaling Technology) was used as a negative control. Lower graph shows quantitative analyses of EGFP-Ku70 intensities at damage foci in time-lapse images of U2OS cells transfected with each siRNA ($n = 5$). **e** Upper panels show that PKC inhibitor (Gö6976), in a dose-dependent manner, rescued the abnormal Ku70 dynamics to DNA damage foci in the presence of HMGB1. Lower graph shows quantitative analyses of EGFP-Ku70 intensities at damaged sites in time-lapse images of U2OS cells ($n = 5$).

MAPK kinase (MEK) and extracellular signal-regulated kinase (ERK) or Camk pathway would explain Ku70 phosphorylation at Ser77/78. Among the two pathways, MAPK pathway is known to mediate HMGB1-TLR4 signaling[54–57].

We further searched for molecules downstream of the HMGB1-TLR4 signaling pathway that affected Ku70 dynamics (Fig. 4c). TLR4 and RAGE share PKC in their downstream pathways[58] and PKC promotes the activation of the MEK-ERK pathway. Therefore, we hypothesized that MAPK pathway downstream of TLR4 might be responsible for the impaired Ku70 dynamics. Small interfering RNAs (siRNAs) against human TLR4 and RAGE, as well as against human MEK and ERK restored the dynamics of Ku70 recruitment to DNA DSB foci in U2OS cells (Fig. 4d), supporting our hypothesis. As a negative control, a control siRNA lacking homology to any mRNA target sequence was shown to not restore the Ku70 recruitment dynamics (Fig. 4d).

PKCα, PKCβ, PKCγ, and PKCε were involved in the MAPK pathway in an integrative manner. Therefore, to test whether these PKC paralogs act upstream of MEK-ERK to affect Ku70 dynamics, we employed a PKC inhibitor (Gö6976) rather than multiple siRNAs (Fig. 4e). As expected, PKC inhibition rescued the delayed EGFP-Ku70 accumulation at DNA damage foci (Fig. 4e).

**PKC is responsible for Ku70 phosphorylation at Ser77/78.** Next, we intended to determine which kinase is responsible for Ku70 phosphorylation at Ser77/78. The Networkin algorithm (http://networkin.info/) and previous reports predicted that PKC might phosphorylate these candidate sites in Ku70 (Fig. 5a). In addition, several other kinases, including ERK, cyclin-dependent kinase 1 (CDK1), ataxia telangiectasia and Rad3-related (ATR), and ataxia telangiectasia mutated (ATM), which have been linked to Ku70[59,60], are also viable candidates for performing the Ku70 phosphorylation (Fig. 5a). Therefore, we performed western blottings of mouse brain tissue samples to examine the levels of the activated forms of several candidate kinases (Supplementary Fig. 6). The result showed that PKCα, PKCβI, PKCβII/δ PKCγ, ATR, MEK, and ERK were activated in 5xFAD mice and recovered by HMGB1 signal interruption (Supplementary Fig. 6), indicating that these kinases remain candidates. Changes in CDK1 and ATM activation were not so remarkable (Supplementary Fig. 6), indicating that they were excluded from candidates.

We mixed various candidate kinases with GST-Ku70 purified by glutathione-Sepharose 4A Fast Flow column, performed in vitro phosphorylation of GST-Ku70 (Supplementary Fig. 6), and confirmed that PKCα and MEK1 kinases could actually phosphorylate Ku70 at Ser77/78 (Fig. 5b). All the results from mass and western blot analyses of active kinases in the cerebral cortex of AD model mice (Figs. 4c and 5b) and analysis of kinases' effects on Ku70 assembly to DNA damage foci (Fig. 4d, e) were consistent with the conclusion of in vitro phosphorylation. Interestingly, the phosphorylated products were cleaved by

contaminated proteinases (Fig. 5b). In the case of PKCα and MEK1 kinase, molecular weight of phosphorylated Ku70 at pSer77/78 was 20 and 30 kDa lower, respectively (Fig. 5b, red and white arrows in left panel), than 100 kDa, the size of non-phosphorylated GST-Ku70 (Fig. 5b, right panel), suggesting that Ku70 was cleaved after phosphorylation at different sites. PKCβI, PKCβII/δ, and PKCγ did not definitely phosphorylate GST-Ku70 (Fig. 5b, lower panel).

In addition, HMGB1 induced Ku70 phosphorylation at Ser77/78 when they were added to the culture medium of U2OS cells at the same concentration (Fig. 5c). HMGB1 at >0.40 nM (monomer with no pre-incubation) induced the similar phosphorylation pattern of PKCα-type cleaved (red and white arrows) as well as full-length (blue arrow) forms of endogenous Ku70 (Fig. 5c, left panel). Pre-incubation to form HMGB1 oligomer[38] inhibited PKCα-type phosphorylation, whereas MEK-type phosphorylation remained (Fig. 5c, left panel), indicating HMGB1 monomer rather than oligomer mainly contributed to Ku70 phosphorylation at Ser77/78. Meanwhile, Aβ monomer did not induce a remarkable change of PKCα-type phosphorylation under the same condition of concentration and incubation time as that of HMGB1 (Fig. 5c, right panel). Similarly, Aβ oligomer/protofibril/fibril forms generated by pre-incubation hardly induced PKCα-type phosphorylation or MEK1-type phosphorylation (Fig. 5c, right panel).

In human postmortem AD brains, western blotting with anti-pSer77/78 antibody revealed the increase of PKCα-type phosphorylation of Ku70 in comparison to non-neurological disease controls (Fig. 5d, left panels) and immunohistochemistry with anti-pSer77/78 antibody suggested that the increase of pSer77/78 Ku70 occurred in cortical neurons, and that pSer77/78 Ku70 together with total Ku70 shifted from the nucleus to the cytoplasm (Fig. 5e). The pattern of bands reactive for anti-Ku70 antibody recognizing C-terminal fragment of Ku70 (Fig. 5d, right panel) was concordant well with our hypothesis that Ku70 is cleaved after PKCα and MEK1 phosphorylation at different sites (Supplementary Fig. 7).

**Interruption of HMGB1 signal suppresses DNA damage and Aβ pathology progression.** The abovementioned analyses indicated that PKCα mainly contributed to the HMGB-mediated Ku70 phosphorylation. In addition, as MEK1 is located in the downstream of PKCα (Fig. 4c), PKCα would be the best target for interrupting the pathological cascade than MEK. To test the therapeutic effect of PKCα inhibition in vivo, we administered Gö6976 intrathecally to 5xFAD mice from 6 weeks of age (Fig. 6a). Gö6976-mediated PKCα inhibition ameliorated DSB detected by γH2AX and 53BP1, markers for DSB, in immunohistochemistry (Fig. 6b) and western blot analyses (Fig. 6c), and it suppressed the pathological phosphorylation of Ku70 at Ser77/78 (Fig. 6c). As expected, Gö6976 reversed the pattern of HMGB1-induced/PKCα-mediated phosphorylation of Ku70 observed in vitro (Fig. 5b, c). Consistently, Gö6976 recovered the alteration rate in

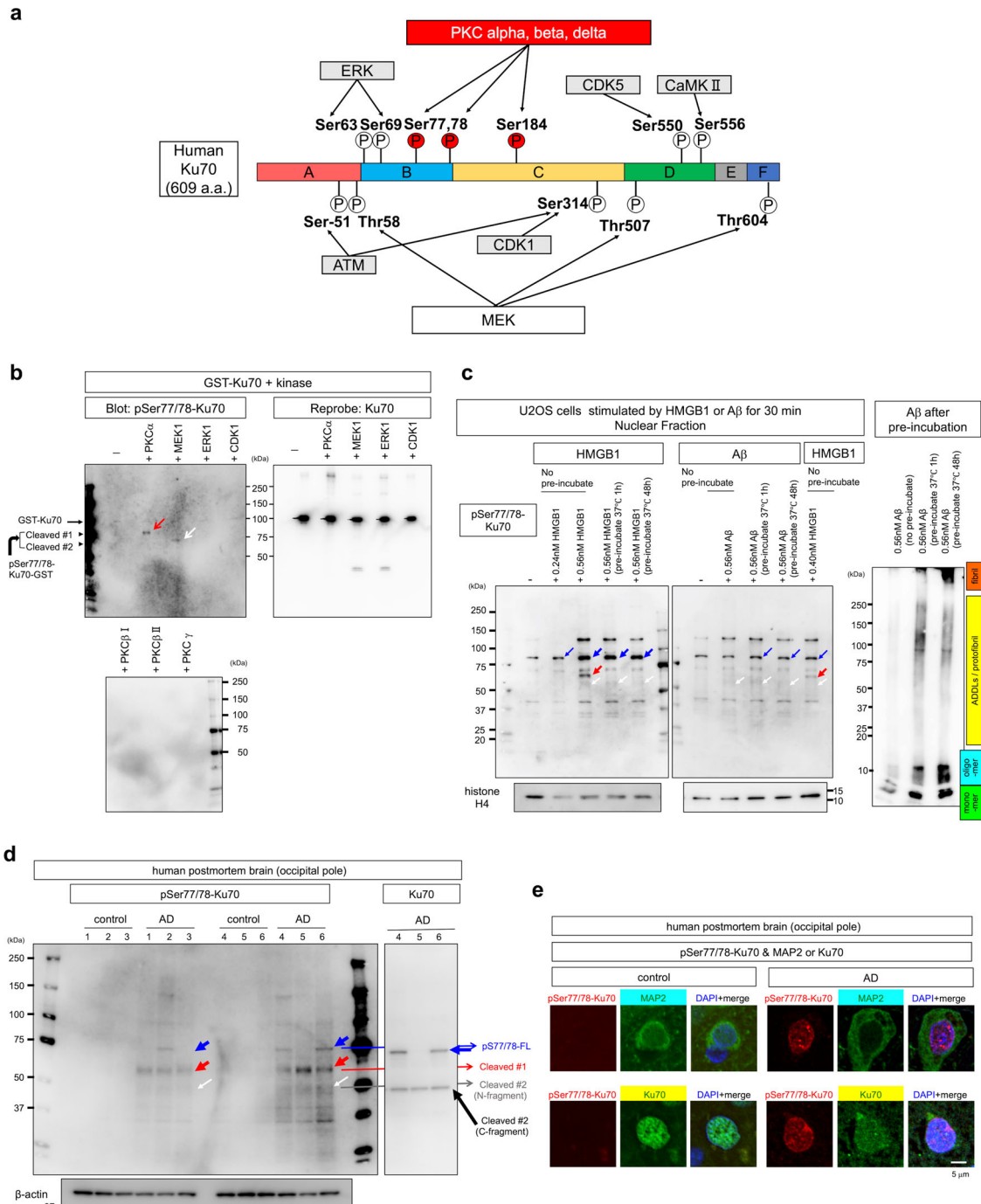

**Fig. 5 PKC is responsible for Ku70 phosphorylation at Ser77/78. a** Predicted kinases responsible for each phosphorylation site of Ku70 based on previous publications[104] and prediction by Networkin, NetPhos, and PhosphoNet. PKCs were the first candidates for Ku70 phosphorylation at Ser77/78. Phosphosites were summarized from previous publication and **b** western blot analysis of anti-pSer77/78 Ku70 antibody revealed the ability of candidate kinases to actually phosphorylate GST-Ku70 at Ser77/78 in vitro. Interestingly, pSer77/78 Ku70 were cleaved at different sites after phosphorylation by PKCα and MEK1. Faint bands at 100 kDa were due to nonspecific interaction with a large amount of GST-Ku70. **c** Comparison of the effects of Aβ vs. HMGB1 on Ku70 phosphorylation at Ser77/78 in U2OS cells. HMGB1 monomer (non-incubated) increased 85 kDa (blue arrow) and the cleaved 65 kDa (red arrow) bands, whereas the cleavage was suppressed in the case of stimulation by HMGB1 oligomer (incubated). The cleaved 55 kDa band was detected both in HMGB1 monomer and oligomer. Aβ monomer, oligomer, protofibril, or fibril did not induce, if any, remarkable increase of 85 and 65 kDa bands reflecting PKCα-type Ku70 phosphorylation. The amount of 55 kDa band (white arrow), MEK-type Ku70 phosphorylation, was also hardly induced by any form of Aβ. **d** Western blot of human postmortem brains (occipital pole) with anti-pSer77/78 antibody. Controls were age-matched non-neurological disease patients. **e** Immunohistochemistry of human postmortem brains (occipital pole) with anti-pSer77/78 antibody.

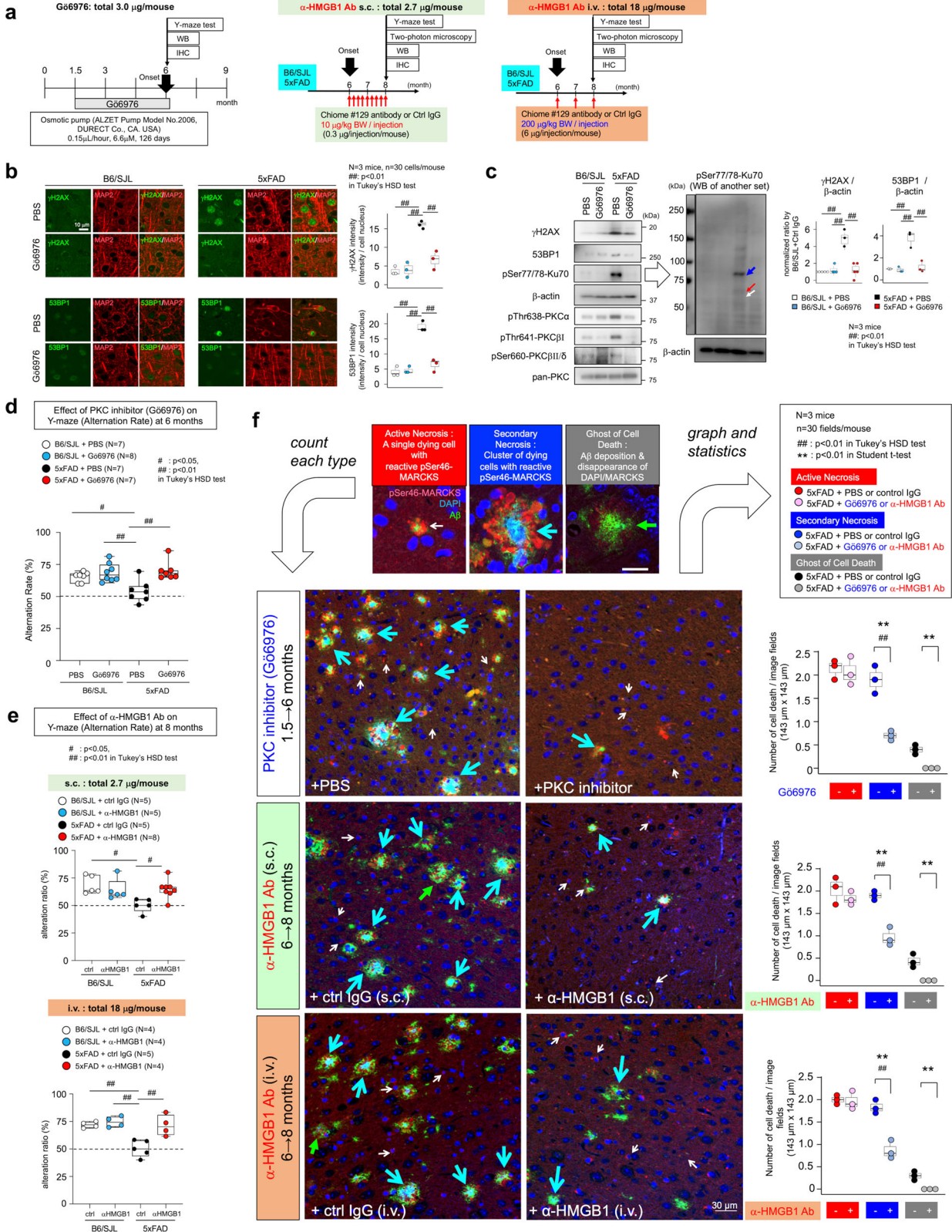

Y-maze tests of 5xFAD mice (Fig. 6d). Interruption of the upstream signal of PKCα activation by antibody #129 similarly recovered cognitive impairment in 5xFAD (Fig. 6e).

As we have recently discovered that Yes-associated protein (YAP)-dependent primary necrosis of cortical neurons at the ultra-early stage of AD[37] is identical to transcriptional repression-induced atypical cell death (TRIAD)[61], and that it is followed by

secondary necrosis of neighboring neurons with unknown characteristics and extracellular Aβ aggregates as ghost of cell death[37], we examined how PKC inhibitor and human mono-clonal anti-HMGB1-Ab #129 affect the two types of necrosis and extracellular Aβ aggregates in AD model mice (Fig. 6f). Interestingly, both treatments reduced the secondary necrosis, a cluster of necrotic neurons surrounding the central deposition of

**Fig. 6 Therapeutic effects of a PKC inhibitor in vivo. a** Experimental protocol. Protocols for human monoclonal anti-HMGB1 antibody (s.c. or i.v.) were similar to those in Supplementary Figs. 4a and 11a. Injection was started after the symptomatic onset of 5xFAD mice. A PKC inhibitor (Gö6976) was continuously injected intrathecally via an Alzet micro-osmotic pump (6.6 µM, 0.15 µL/h) from 1.5 to 6 months of age. Y-maze tests were performed at 6 months of age and brain samples were dissected. **b** Immunohistochemistry of mice treated by the protocol in **a**. γH2AX and 53BP1, the markers of DSB, revealed that DNA damage increased in 5xFAD mice was recovered by PKC inhibitor (Gö6976). **c** Left panels: western blot analysis of Gö6976-treated and non-treated mice confirmed the increase of DNA damage in 5xFAD mice and the recovery by Gö6976. Western blotting of various PKCs confirmed the activation of PKCα and the increase of pSer77/78 Ku70 in the cortex of 5xFAD mice. Gö6976 treatment similarly suppressed PKCα and pSer77/78 Ku70 in the cortex of 5xFAD mice. Middle panels: full-scan of another set of samples reconfirmed the increase of pSer77/78 Ku70, which matched the pattern of PKCα-mediated Ku70 phosphorylation (Fig. 4a, b). Right graphs show quantitative analysis of the γH2AX and 53BP1 band signals. **d** Gö6976 treatment (shown in **a**) rescued cognitive impairment of 5xFAD mice in Y-maze test alteration ratio. **e** Human monoclonal anti-HMGB1 antibody also recovered cognitive impairment of 5xFAD mice in Y-maze test alteration ratio. **f** Rescuing effect of Gö6976 and human monoclonal anti-HMGB1 antibody on the secondary necrosis in 5xFAD mice. Left panels show the immunohistochemistry with anti-pSer46-MARCKS antibody that differentiate active necrosis, secondary necrosis, and Aβ extracellular aggregates reflecting the ghost of neuronal necrosis[37]. Right graphs show quantitative analysis in each group of active necrosis, secondary necrosis, and Aβ-positive ghost of cell death (Student's t-test). Active necrosis was not significantly changed, whereas secondary necrosis and the following ghost of cell death were decreased. Especially, secondary necrosis was remarkably decreased in multiple-group comparison (Tukey's HSD test). Box plots show the median, quartiles, and whiskers that represent data outside 25th to 75th percentile range.

Aβ, and extracellular Aβ aggregates, but did not basically affect the primary active necrosis, an original necrosis of a single neuron with intracellular deposition of Aβ (Fig. 6f).

In addition, we showed that HMGB1 stimulation of normal human induced pluripotent stem cell (iPSC)-derived neuron induces endoplasmic reticulum (ER) ballooning (Fig. 7a–c), which was similar to the TRIAD observed in AD-iPSC-derived neurons[37]. Human monoclonal anti-HMGB1-Ab #129 and PKC inhibitor suppressed the frequency of ER ballooning (Fig. 7a–c).

**TRIAD is the cell death phenotype of senescence**. Senescent glial and neuronal cells have been identified in neurodegeneration and the aging brain[30–35], and senescence shared multiple features such as DNA damage, SASP/DAMP secretion (especially secretion of HMGB1), transcriptional change, and apoptosis resistance[26] with TRIAD necrosis[37,38,61–63].

Therefore, we tested the identity between TRIAD and senescence. A nuclear membrane protein Lamin B1, as a senescence marker, generated the ring-like stains in normal neurons, astrocytes, and microglia of C57BL/6 mice (B6) (Supplementary Fig. 8a, white arrow). The ring of Lamin B1 became unclear or disappeared in a part of neurons and astrocytes of 5xFAD or APP-KI mice (Supplementary Fig. 8a, yellow arrow). The Lamin B1 ring included abnormal foci, whereas the ring by itself mostly remained in the microglia of 5xFAD or APP-KI mice (Supplementary Fig. 8a, yellow arrow).

Triple staining of YAP, pSer46-MARCKS, and Lamin B1 revealed that Lamin B1 ring-negative senescent cells are frequently nuclear YAP negative, indicating the overlap of TRIAD and senescence phenotypes (Fig. 8a, thin white arrow). The TRIAD-senescent cells surrounded the larger pSer46-MARCKS-positive focus of primary TRIAD necrosis (Fig. 8a, thick white arrow). On the other hand, nuclear YAP-positive cells kept Lamin B1 ring (Fig. 8a, green arrow). Intravenous administration of human monoclonal anti-HMGB1-Ab (#129) obviously reduced the number of senescent cells, both in the brain of 5xFAD and APP-KI mice (Fig. 8b). Intravenous administration of human monoclonal anti-HMGB1-Ab #129 rescued senescence of neurons, astrocytes, and microglia (Supplementary Fig. 8b).

We further addressed senescence-related inflammation around glial cells and the therapeutic effect of human monoclonal anti-HMGB1-Ab (#129) on the inflammation (Fig. 9). Nuclear translocation of nuclear factor-κB was increased in 5xFAD and APP-KI mice, which was suppressed by anti-HMGB1-Ab (Fig. 9a).

Monocyte chemotactic protein 1 (MCP1) and Interleukin-6 (IL-6) around the astrocyte or microglia were increased in 5xFAD

and APP-KI mice, which were also suppressed by anti-HMGB1-Ab (Fig. 9b).

These data further supported the idea that TRIAD is the cell death phenotype of senescence, and that anti-HMGB1-Ab suppressed senescence-associated glial activation/secretion.

Immunohistochemistry with specific markers for TRIAD and other types of cell death including pyroptosis, paraptosis, and necroptosis revealed that the late-onset necrosis in the cerebral cortex of 5xFAD and APP-KI mice was almost exclusively TRIAD (Supplementary Fig. 9). The dominancy of TRIAD to the other types of necrosis was also confirmed with regards to secondary necrosis around extracellular Aβ aggregates (Supplementary Fig. 10).

**HMGB1 signal impairs neuronal primary cilia via DNA-damage-induced change of transcriptional expression profiles**. The accumulation of DNA damage due to impaired DNA repair resulting from Ku70 phosphorylation at Ser77/78 might alter the profile of gene expression. We compared gene expression profiles by RNA sequencing of control neurons, amyloid precursor protein (APP) mutant neurons with homozygous APP gene mutations (KM670/671NL), HMGB1-treated APP mutant neurons, and antibody #129-treated APP mutant neurons that were differentiated from human normal or mutant iPSC cells (iPSC neurons) generated by genome editing[64]. We examined 401 genes with expression profiles that differed across these treatment types (Fig. 10a). Genes with altered expression profiles were clustered by Gene Ontology (GO) into five groups (G1–G5). Fifty-six genes were identified in 5 GO groups and 26 GO terms were significantly enriched (Fig. 10b). In parallel, we examined PPI networks by inputting the 56 genes into STRING version 11.0 (https://string-db.org/) and found 5 interaction network groups (S1–S4) and a group of unlinked genes (Others) (Fig. 10c). Comparison of the GO and STRING groupings revealed frequent overlaps of member genes (Fig. 10d) and good correlation between the two different categorization methods (Fig. 10e). Intriguingly, five genes related to neuronal primary cilia were identified in overlapping groups (S1, G4). Notably, the expression of DNAAF4 (Dynein Axonemal Assembly Factor 4), mutations and translocations in which have been associated with deficits in reading and writing[65–67], was downregulated by HMGB1 in AD-iPSC neurons, whereas components of the cilia, such as CCDC114 (coiled-coil domain-containing protein 114), and DNA2 (DNA replication ATP-dependent helicase/nuclease DNA2) were upregulated.

Neuronal primary cilia have been implicated in proliferation, differentiation, and migration of neural stem progenitor cell and

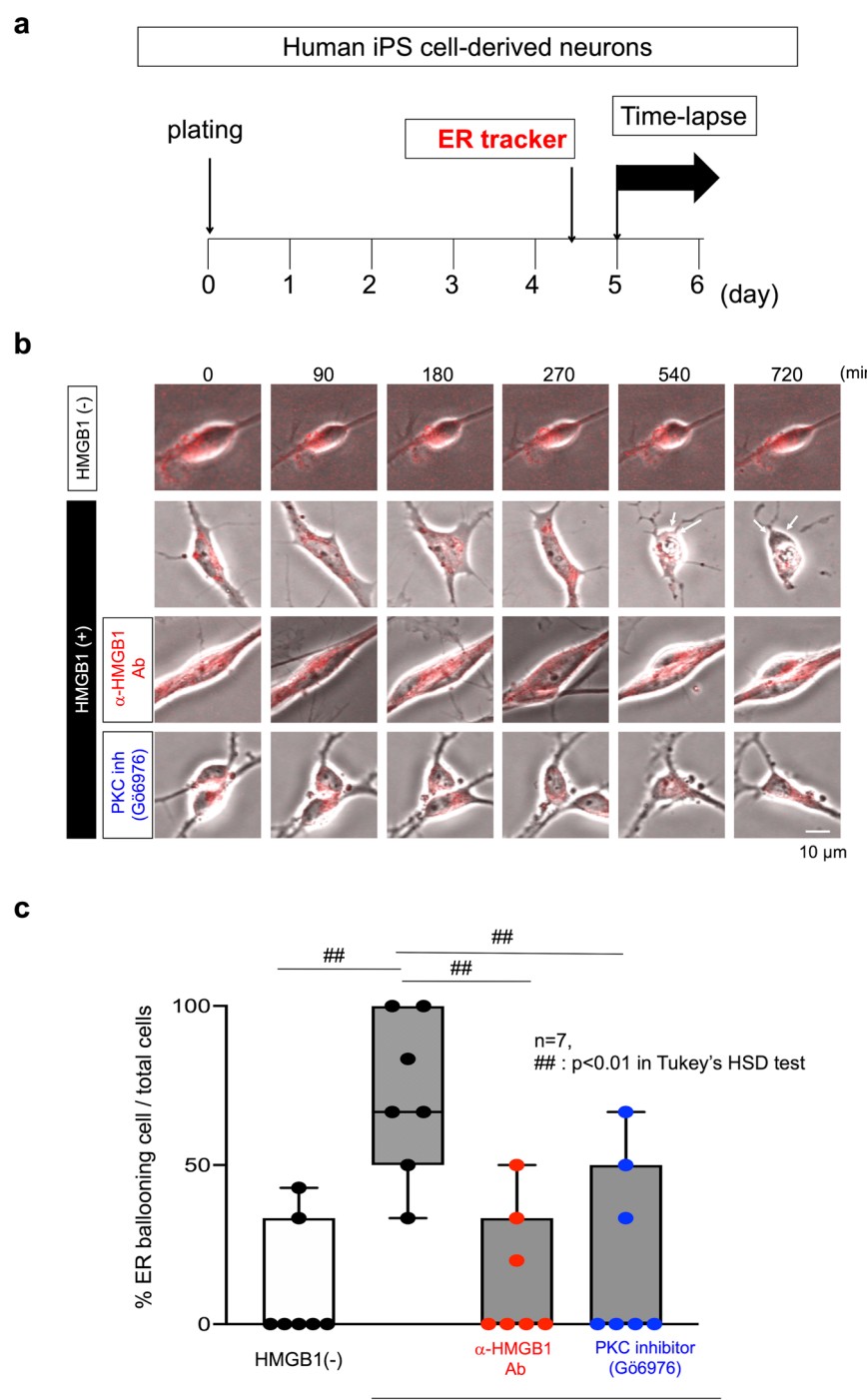

**Fig. 7 HMGB1-induced TRIAD necrosis of iPSC-derived neurons and the suppression by human monoclonal anti-HMGB1 antibody. a** Time-lapse imaging protocol used in **b** and **c**. HMGB1 antibody or PKC inhibitor was added to the medium of iPSC-derived neuron simultaneously with HMGB1. **b**, **c** Time-lapse imaging of normal iPSC-derived neurons. Addition of HMGB1 increased ER ballooning, which was suppressed by the HMGB1 antibody or PKC inhibitor. Box plots show the median, quartiles, and whiskers that represent data outside the 25th to 75th percentile range.

differentiated neurons[68]. Mutations of cilia-associated genes in human patients produce a variety of phenotypes including microcephaly, cerebellar hypoplasia, and cognitive impairment[69]. In addition to the role of neuronal primary cilia in the development of the brain and neural network[68–70], postnatal ablation of cilia in adult neurogenesis impairs learning and memory in adulthood[71], although knowledge of the roles of neuronal primary cilia in neurodegeneration is limited[72,73]. Therefore, human iPSC neurons were stained and their frequency

and length were quantified. Blind analyses revealed a decrease and shortening of neuronal primary cilia in *APP* mutant iPSC neurons (Fig. 10f). Moreover, we observed a small number of neuronal primary cilia in the cerebral cortex of the non-diseased human brains, but very few in postmortem human AD brains (Fig. 10g). Impairment of neuronal primary cilia may not be linked to neuronal cell death but could be causative for neuronal dysfunction and cognitive decline downstream of the HMGB1-TLR4-PKC-Ku70 pathway, in parallel with neuronal necrosis.

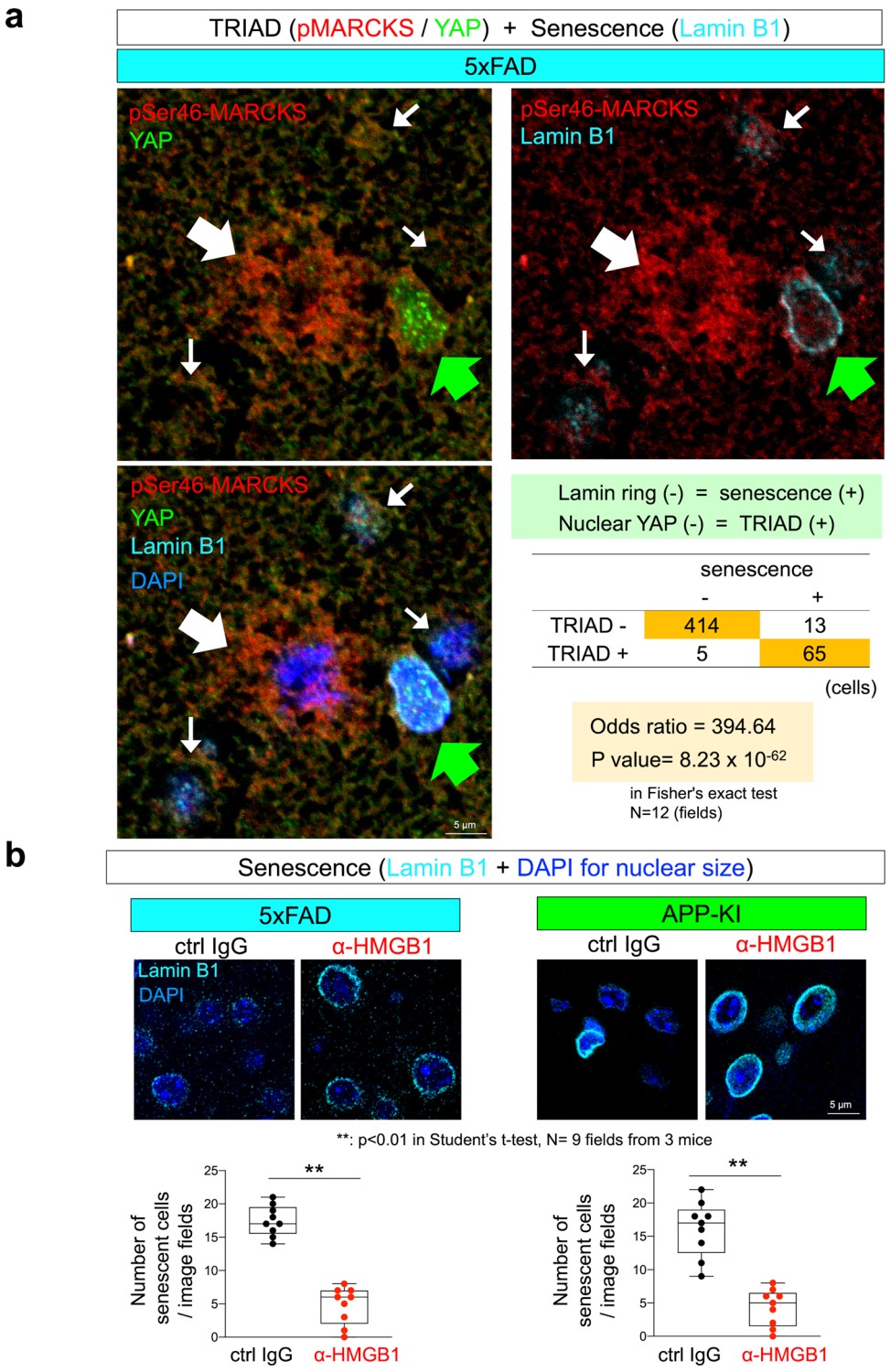

**Fig. 8 Overlap of TRIAD and senescence phenotypes in neurons of AD model mice. a** Triple staining of the cerebral cortex of AD model mice was performed at 8 months of age. A Lamin B1 ring-negative and nuclear YAP-negative cell (TRIAD-senescence overlapped phenotype) was observed at the center of pSer46-MARCKS stains (thick white arrow). Around the pSer46-MARCKS foci, many Lamin B1 ring-negative and nuclear YAP-negative cells are observed, showing overlap of TRIAD and senescence phenotypes (thin white arrow). Nuclear YAP-positive cells kept Lamin B1 ring (green arrow). The right lower table shows a positive relationship between TRIAD and senescence, and the statistic confirmation by Fisher's exact test. **b** Intravenous administration of human monoclonal anti-HMGB1 antibody (#129) reduced the number of senescent cells both in the cerebral cortex of 5xFAD and APP-KI mice.

**Interruption of HMGB1 signal recovers synapse.** Subcutaneous or intravenous injection of human monoclonal anti-HMGB1-Ab #129 into 5xFAD mice (Supplementary Fig. 11a) suppressed MARCKS phosphorylation at Ser46 in immunohistochemistry and western blot analyses (Supplementary Fig. 11b, c), recovered decrease of dendritic spines (Supplementary Fig. 11d), and recovered cognitive impairment (Fig. 6e). Although mouse anti-HMGB1-Ab did not significantly alter the number and area of the Aβ plaques[38], human antibody #129 suppressed extracellular Aβ aggregates (Supplementary Fig. 11e). Similarly, in APP-KI mice

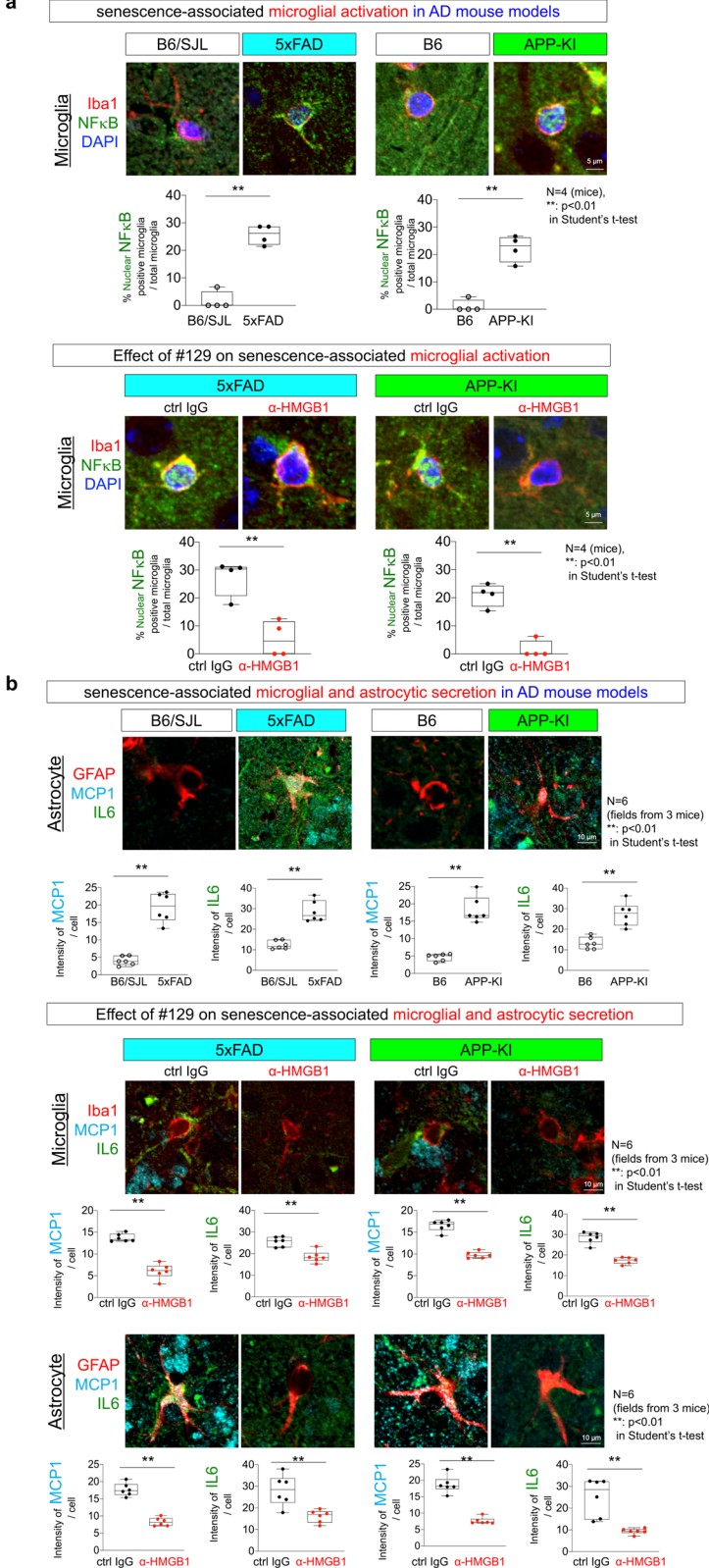

**Fig. 9 Suppression of glial senescence by anti-HMGB1 antibody. a** The ratio of activated microglia in which NFκB were translocated to the nucleus was increased in 5xFAD and APP-KI mice at 8 months of age (upper panels). Microglia in the occipital cortex were counted in a visual field of a slide made from each mouse ($N = 4$). Activated microglia with nuclear NFκB were suppressed in 5xFAD and APP-KI mice by intravenous injection of human monoclonal anti-HMGB1 antibody (#129) according to the protocol in Fig. 6 (lower panels). **b** Secretion of SASP (MCP1 and IL-6) was increased in 5xFAD and APP-KI mice at 8 months of age (upper panels). Signal intensities of MCP1 and IL-6 were acquired in the circular area of 50 µm diameter surrounding a cell. The average value was calculated with their signal intensities of multiple circular areas in two visual fields derived from each mouse ($N = 3$). Secretion of SASP was suppressed in 5xFAD and APP-KI mice by intravenous injection of human monoclonal anti-HMGB1 antibody (#129) according to the protocol in Fig. 6.

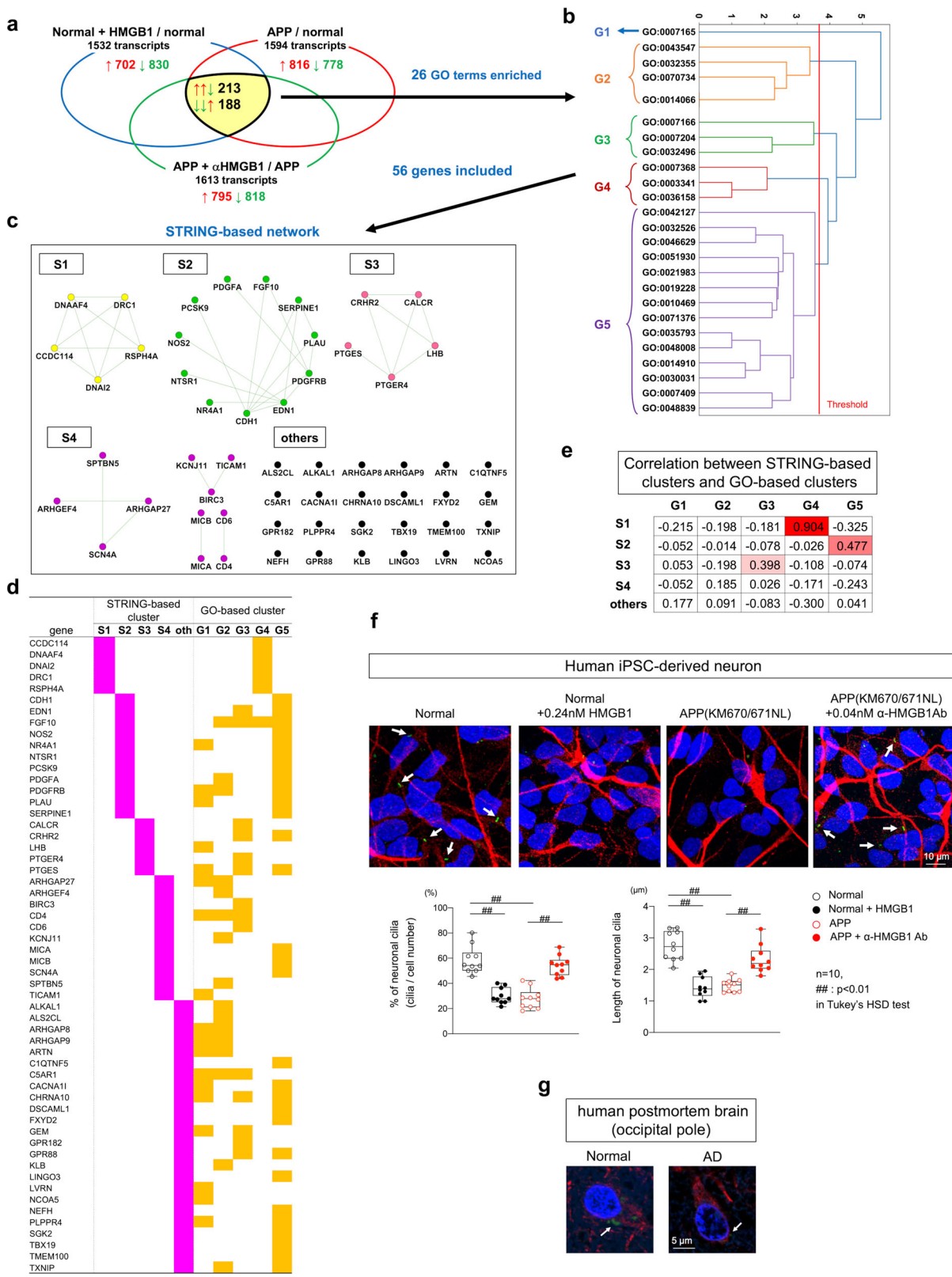

subcutaneous or intravenous injection of human antibody #129 (Supplementary Fig. 11f) recovered MARCKS phosphorylation at Ser46 (Supplementary Fig. 11g), decrease of dendritic spines (Supplementary Fig. 11h), cognitive impairment (Supplementary Fig. 11i), and extracellular Aβ aggregates (Supplementary Fig. 11j).

Among the list of significantly changed proteins after treatment of 5xFAD with HMGB1-Ab, the increases in Tau and Syn1

protein phosphorylation were observed with statistical significance (Fig. 4a, b). Ectopic pSer203-tau in the downstream of MAPK pathway induces synapse instability from the early stages of frontotemporal lobar degeneration (FTLD) pathology[74]. Therefore, we examined pSer203-tau in AD pathology (Supplementary Fig. 12). Immunohistochemistry of 5xFAD mice after subcutaneous or intravenous injections of human control IgG

**Fig. 10 HMGB1 signaling impairs neuronal primary cilia via DNA damage-induced changes in transcription profiles. a** Expression profile changes in AD pathology following extracellular HMGB1 treatment by RNA sequencing analysis. Pan-neurons were differentiated from normal human induced pluripotent stem cells (iPSCs) or from mutant iPSCs carrying heterozygous APP mutations (KM670/671NL) generated by genome editing, treated with HMGB1 and/ or HMGB1 antibody addition to the medium. Venn diagrams show common changes in gene expression shared by HMGB1 treatment and AD pathology, or the effect of anti-HMGB1 antibody treatment on gene expression in AD pathology. Detailed experimental procedures are described in the "Methods". **b** GO-based cluster analysis of 401 candidate genes with gene expression altered by HMGB1 in AD pathology. Fifty-six genes were included and GO terms were clustered into five GO-based groups where 26 GO terms were significantly concentrated. Dendrogram of Hierarchical Clustering was constructed based on the Ward's criterion. **c** PPI network analysis of the 56 genes identified in **b** by String version 11.0 (https://string-db.org/). Four interaction networks were identified (S1–S4), with the remaining gene products labeled as "others". **d** Comparison between the GO-based and STRING-based groupings described in **b** and **c**. **e** Correlation between the two groups described in **b** and **c**. Numbers represent Pearson's correlation coefficients. **f** Analyses of neuronal primary cilia in normal and heterozygous APP mutant human iPSC neurons. Representative images are shown in the panels. Frequency and length of neuronal primary cilia were quantitatively analyzed by Imaris and are shown in the graph. **g** Decrease in neuronal primary cilia in postmortem human AD brains.

revealed a dot-like pattern of accumulation of pSer203-tau in the occipital cortex neuropil of 5xFAD mice at 8 months of age (Supplementary Fig. 12a). These dots were co-stained with PSD95 (Supplementary Fig. 12a), similar to the ectopic localization of pSer203-tau identified in mouse and human FTLD pathologies[74]. Subcutaneous or intravenous injections of human monoclonal anti-HMGB1-Ab (#129) reduced the frequency of pSer203-tau co-stained with PSD95 (Supplementary Fig. 12a). Western blot analysis confirmed the increase of pSer203-tau in 5xFAD mice and its amelioration by antibody #129 (Supplementary Fig. 12b). We also confirmed the increased pSer203-tau abundance and its ectopic localization to dendritic spines in postmortem parietal cortex samples from human AD patients (Supplementary Fig. 12c). No side effects were observed regarding external appearance, body weight gain during aging, and pathological examination of various systemic organs (Supplementary Fig. 13).

## Discussion
This study reveals that extracellular HMGB1 triggers phosphorylation signaling to Ku70, decreases Ku70 affinity to DNA damage foci, and impairs the NHEJ function of Ku70 in next recipient cells. Consequently, DNA damage is accumulated in neurons receiving HMGB1 via TLR4. The linkage of extracellular DAMPs and impairment of DNA damage repair in recipient cells has not been reported. TLR4 signaling has not been implicated in such impairment of DNA damage repair. Meanwhile, some previous data suggested that deficiency of DAMPs essential for genome stability such as HMGB1 would impair DNA damage repair in dying cells[75]. However, the mechanism revealed by this study is obviously distinguished from the previous one in the aspect of cells in which impairment of DNA damage repair occurs: the cell releasing HMGB1 (donor cells) or the cell receiving HMGB1 via TLR4 (recipient cells).

This study touches controversial relationships between PKC activity and AD pathology. PKC hypofunction and therapeutic application of PKC activator were once suggested[76,77], whereas clinical trial of a PKC activator Bryostatin 1 for human AD patients was unsuccessful (https://clinicaltrials.gov/ct2/show/NCT02221947?term=PKC%2C+alzheimer&draw=2&rank=1). Instead, administration of Bryostatin 1 worsened cognitive symptoms of the treated AD patients (https://clinicaltrials.gov/ct2/show/results/NCT02221947?term=PKC%2C+alzheimer&draw=1&rank=1). On the other hand, a large-scale genetic study of late-onset AD (LOAD) families revealed that hyperactivity of PKCα is a risk factor of AD[78]. Another group reported that PKCδ elevated in AD may increase the expression of BACE1[79]. However, the result could not explain the meaning of active form mutations of PKCα, a distinct gene from PKCδ. Although this study supported the increase of PKCδ, the activation of PKCδ was not suppressed by HMGB1-Ab

(Supplementary Fig. 6). Moreover, their hypothesis based on BACE1 is distinct from our Ku70-mediated mechanism.

We show that PKCα directly phosphorylates Ku70 and plays a major role in the impairment of DNA damage repair mechanism. Even though PKCs compose an essential group of kinases for memory formation under the physiological condition[80–83], our data well explains the result that hypermorph mutations of PKCα are a risk factor of LOAD[78]. A certain difference in the action mode of PKCα, such as extent and time constant of PKCα hyperactivity or collaborating factors with PKCα, might determine whether hyperactive PKCα leads to beneficial or deteriorating outcomes in the AD pathology.

In addition, we reveal the upstream mechanisms how activation of PKCα impairs Ku70 function via competition between 14-3-3 binding to phosphorylated Ser77/78 and SIRT1 that deacetylates Lys331 (Fig. 2d). Recruitment and binding of Ku70 to DNA damage foci were both impaired by abnormal phosphorylation at Ser77/78 and acetylation at Lys331 of Ku70 (Figs. 1 and 2). Two downstream pathways are also revealed from abnormal phosphorylation of Ku70 to *trans*-neuronal propagation of necrosis and to impairment of neuronal primary cilia via transcription.

This study provides a new concept that HMGB1 is the key molecule to propagate DNA damage and secondary necrosis, which has not been reported as far as we know. The hypothetical concept accords well with our previous observations that intracellular Aβ deposition sequesters YAP, a co-activator of the transcription factor TEAD essential for cell survival, and induces the primary neuronal necrosis[37]. Secondary necrosis occurs in neurons around primary necrosis and increases the size of extracellular Aβ aggregates[37]. If necrosis occurs in multiple neurons stochastically, multiple neurons cannot form such a cluster of secondary necrosis. Hence, the concept of HMGB1-mediated propagation of DNA damage and TRIAD necrosis (Supplementary Fig. 14) can solve the enigma of stochastic disequilibrium of necrosis. Interestingly, dynamic molecular network analysis of big data from four AD and four FTLD mouse models predicted the HMGB1-TLR4 signal as the core pathway commonly shared by AD and FTLD pathologies[84].

HMGB1 is also released from hyperactive neurons[38]. HMGB1 secretion explains well the previously described phenomenon that human cortical neurons of the default mode network are most vulnerable in AD pathology[85]. HMGB1 secretion also explains well that neuronal hyperactivity is correlated to the extent of DNA damage under AD pathology[19,20].

The second concept obtained from this study is identification of propagated necrosis to be TRIAD. Previous studies, although the detailed mechanisms have not been revealed, suggested that DNA damage might induce necrosis[86,87], and that p53 contributes to non-apoptotic death of developmental and mature neurons[88]. Considering that p53 is suppressed under ER stress[89],

DNA damage-induced necrosis in p53-deficient cells reported by the Thompson and colleagues[86] is highly interesting in regards of homologous morphologies to TRIAD necrosis[61]. DNA damage-induced necrosis in p53-deficient cells and DNA damage-induced necrosis in AD identified in this study could be identical and defined as TRIAD necrosis[61]. Moreover, morphological similarity between TRIAD and paraptosis[90] is also noteworthy. However, paraptosis is suppressed by transcriptional repression in contrast to TRIAD accelerated by transcriptional repression[61,90] and the exact relationship between the two types of necrosis should be clarified by further investigation in the future for unifying DNA damage-induced necrosis. Our analysis in this study also reveals the discrepant frequency of TRIAD and paraptosis in human and mouse AD brains on the basis of their markers. The analogous predominance of TRIAD was also discovered in multiple FTLD pathologies[91], suggesting the generality in neurodegenerative diseases and specificity in senescence of TRIAD type of necrosis.

Moreover, increased DNA damage by Ku70 phosphorylation widely influences gene expression profiles, including specific transcriptional changes in genes related to neuronal primary cilia. Although fragmented knowledge exists regarding the role of neuronal primary cilia in neurodegenerative disease[72,73] or in AD[92], the whole scheme remains largely unknown. This study presented a new scheme, to the best of our knowledge, that in the downstream of extracellular HMGB-induced signal, neuronal primary cilia are functionally impaired via gene expression. In addition to necrosis propagation, the propagation of neuronal dysfunction by HMGB1 may also occur in neuronal primary cilia.

The third concept suggested by this study is the similarity between senescence and TRIAD. Both biological processes share DNA damage, SASP/DAMP secretion (especially secretion of HMGB1), transcriptional change, and apoptosis resistance[26] with TRIAD necrosis[37,38,61–63,91]. Biomarker-based analyses in this study further supported the identity of TRIAD and senescence, indicating that TRIAD is the end stage of senescence phenotype of neurons in neurodegeneration and aging brains.

Finally, the reason why this system works in neurodegenerative diseases and brain aging remains unknown. DAMP signals play an essential role in protection against infectious organisms such as parasite and virus. It is possible that this system malfunctions in endogenous parasite/virus mimicry, i.e., accumulation of disease proteins in the neurons and glia. This idea might be too speculative but worth pursuing in the future.

## Methods

**AD mouse models**. 5xFAD transgenic mice overexpressing mutant human *APP* (770) with the Swedish (KM670/671NL), Florida (I716V), and London (V717I) familial AD (FAD) mutations and human PS1 with two FAD mutations (M146L and L285V) were purchased from The Jackson Laboratory (CA, USA). Both the *APP* and *PS1* transgenes were under the control of the mouse *Thy1* promoter[93]. The background of the mice was C57BL/SJL, which was generated by crossbreeding C57BL/6J female and SJL/J male mice. A comprehensive proteomics analysis was performed with brain tissue samples from male 5xFAD mice at 1, 3, 6, and 12 months of age, as previously described[40]. APP-KI ($App^{NL-G-F/NL-G-F}$) mice[94] carry Swedish (KM670/671NL), Beyreuther/Iberian (I716F), and Arctic (E693G) mutations in humanized *APP* mouse gene. PKC inhibitor (Go6976, 6.6 µM) was administered to the subarachnoid space of mouse cerebral cortex at 0.15 µl/h from 1.5 to 6 months of age by using osmotic pump (ALZET Osmotic Pumps #2006, Cupertino, CA, USA).

**Human AD brains**. Brain samples for proteome analysis were dissected from AD and normal disease control bodies, and were deep frozen (−80 °C) within 1 h after death. Their pathological diagnoses were determined based on immunohistochemistry. AD brains used for this study did not include other pathological changes, such as Lewy bodies, TDP43 cytoplasmic aggregates, or argyrophilic grains. Temporal pole and occipital pole tissues were dissected from five brains from each group. Control brains were derived from age-matched patients who died due to non-neurological diseases.

**Nuclear extract preparation from U2OS cells**. U2OS cells were washed three times with phosphate-buffered saline (PBS), collected by a cell scraper, and centrifuged at 1000 × g for 5 min at 4 °C. The pellet was resuspended in 8× vol of lysis buffer (20 mM Hepes pH 7.9, 1 mM dithiothreitol (DTT), 1 mM EDTA, 10% Glycerol, 0.5 mM spermidine, and 0.5% protease inhibitor cocktail [Calbiochem, San Diego, CA, USA]) with 0.3% Nonidet P-40, placed on ice for 5 min, and centrifuged at 15,000 × g for 10 min at 4 °C. The pellet was suspended with 1× vol. of the lysis buffer, added with 1× vol. of 2 M KCl (final concentration: 1 M KCl), mixed gently, placed on ice for 30 min, and centrifuged at 100,000 × g for 30 min at 4 °C. The supernatant was used for proteomics analysis or SDS-polyacrylamide gel electrophoresis (SDS-PAGE).

**Mass spectrometry**. Comprehensive proteomics analyses were performed using brain tissue samples from male 5xFAD and littermate control (non-transgenic) mice at 1, 3, 6, and 12 months of age, or by using human postmortem brain tissues (occipital and temporal poles) of AD patients and non-neurological disease control, as previously described[40]. Briefly, brain extracts were denatured via detergent and heat treatment, reduced to block cysteine bonds, and digested with trypsin. The phosphopeptides were enriched using the Titansphere Phos-TiO kit (GL Sciences, Inc., Tokyo, Japan) and labeled using an iTRAQ Reagent multiplex kit (SCIEX, Inc., MA, USA). Liquid chromatography using a strong cation exchange column was used to separate the enriched and labeled phosphopeptides. Each fraction was separated using a DiNa Nano-Flow LC system (Eksigent NanoLC-Ultra 1D Plus system, SCIEX, Inc., Tokyo, Japan) and 0.1 × 100 mm C18 columns (KYA Technologies Corporation, Tokyo, Japan). The ion spray voltage applied to the samples in the NanoLC-connected Triple TOF 5600 System (SCIEX, Inc., Tokyo, Japan) was 2.3 kV. The information-dependent acquisition setting was 400–1250 *m/z*. Proteomics analysis of nuclear extract from U2OS cells was performed similarly but without enrichment of phosphopeptides.

**Modeling of Ku70 phosphorylated at Ser77 and Ser78**. Structure of Ku70 phosphorylated at Ser77 and Ser78 was not available at the PDB. Therefore, building tools of PyMOL (Shrödinger, LLC) were used to model the phosphorylated Ku70 based on the wild-type structure (PDB ID: 1JEY). The phosphate groups were placed in standard positions of the modeled Ku70.

**Molecular dynamics**. MD simulations were performed using Gromacs package 2018.1[95]. The structure of Ku77 and Ku78 complex was taken from the PDB with the number 1JEY and its missing residues (223–230 of Ku70 and 171–180 of Ku80) were modeled using CHARM-GUI[96–98]. The CHARMM36 force field was used to describe the system. After energy minimization and equilibration, MD production runs were performed at 303.15 K in a rectangular box with 1.0 nm edge distance, which were filled with water molecules and neutralized with 0.15 M NaCl. MD production runs were terminated at 30 ns. Distances between O atom of Ser77-Ku70 and P or O of DNA were calculated using the data sets of MD trajectories.

**Plasmid construction**. Human pEGFP-C1-FLAG-Ku70 (EGFP-Ku70-WT) plasmid was purchased from Addgene (#46957, Watertown, MA, USA), from which phospho-mimetic or dephospho-mimetic mutants were generated. The primer sets for S77A, S77D, S77E, SS77/78AA, SS77/78DD, and SS77/78EE mutants were as follows. S77A-forward: 5′-CATAGCCAGTGATCGAGATCTCTTGGC-3′ and S77A-reverse: 5′-CGATCACTGGCTATGATCTTACTGATG-3′, S77D-forward: 5′-CATAGACAGTGATCGAGATCTCTTGGC-3′ and S77D-reverse: 5′-CGATCACTGTCTATGATCTTACTGATG-3′, S77E-forward: 5′-CATAGAAAGTGATCGAGATCTCTTGGC-3′ and S77E-reverse: 5′-CGATCACTTTCTATGATCTTACTGATG-3′, SS77/78AA-forward: 5′-CATAGCCGCTGATCGAGATCTCTTGGC-3′ and SS77/78AA-reverse: 5′-CGATCAGCGGCTATGATCTTACTGATG-3′, SS77/78DD-forward: 5′-CATAGACGATGATCGAGATCTCTTGGC-3′ and SS77/78DD-reverse: 5′-CGATCATCGTCTATGATCTTACTGATG-3′, SS77/78EE-forward: 5′-CATAGAAGAAGATCGAGATCTCTTGGC-3′ and SS77/78EE-reverse: 5′-CGATCTTCTTCTATGATCTTACTGATG-3′.

To construct pGEX-6P-1-Ku70-WT, -S77A, -S77D, -S77E, -SS77/78AA, -SS77/78DD, and -SS77/78EE plasmids, human Ku70 wild-type, phospho-mimetic, and dephospho-mimetic Ku70 cDNAs were amplified, respectively, from pEGFP-C1-FLAG-Ku70-WT, S77A, S77D, S77E, SS77/78AA, SS77/78DD, and SS77/78EE plasmids using the primers 5′-ATGCGGATCCTCAGGGTGGGAGTCATATTAC-3′ and 5′-ATGCCTCGAGTTAGTCCTGGAAGTGCTTGG-3′. After digestion with BamHI and XhoI, human Ku70 cDNA fragments were subcloned into pGEX-6P-1 (Cytiva, Marlborough, MA, USA).

**GST-fusion proteins**. Rosetta™(DE3) competent cells (70954, Merck, Darmstadt, Germany) were transformed with pGEX-6P-1-Ku70-WT, -S77A, -S77D, -S77E, -SS77/78AA, -SS77/78DD, or -SS77/78EE plasmid, respectively, grown until OD600 reached to 0.4, and incubated with 1 mM isopropyl β-D-1-thiogalactopyranoside at 37 °C for 2 h. The cells were collected, suspended in lysis buffer (0.1% Triton X-100, 0.1% Lysozyme, 1 mM DTT, protease inhibitor cocktail (539134, Calbiochem, San Diego, CA, USA)), and sonicated ten times on ice. Glutathione S-transferase (GST)-fusion proteins were purified by glutathione-sepharose 4B (17075601, Cytiva, Marlborough, MA, USA) and eluted with 2 ml of GST elution

buffer (10 mM Glutathione, 0.1% Triton X-100 pH 8.0, and protease inhibitor cocktail (539134, Calbiochem, San Diego, CA, USA)).

**Electrophoretic mobility gel shift assay**. Sense and antisense oligonucleotides, 5′-TAGAGACGGGG TTTCACCGTGTTACCAGG-3′ and 5′-GATCTAACACAC GGTGAAACCCC GTCTCTATGC-3′, were annealed by stepwise cooling down from 95 °C to room temperature (RT) and were labeled with digoxigenin (DIG) using the DIG Gel Shift Kit (3353591910, Roche, Basel, Switzerland) to generate DIG-labeled probes for detecting Ku70 bound to DNA DSB[99]. The DIG-labeled probe (0.8 ng) was incubated with 40 μg of GST-fusion proteins for 30 min at RT in 25 μL of binding buffer (1 μg of poly[d-(I-C)], 0.1 μg of poly-L-lysine, 10 mM Tris-HCl pH 7.5, 50 mM NaCl, 1 mM MgCl₂, 0.5 mM DTT, 4% glycerol)[99]. The samples were separated by electrophoresis with 8% native polyacrylamide gel and 0.5× TBE buffer at 120 V for 1 h at 4 °C. After electrophoresis, the protein-probe complex and probe were transferred to Amersham Hybond-N+ membranes (RPN303B, Cytiva, Marlborough, MA, USA) for 30 min at 400 mA and cross-linked at 120 mJ for 3 min. The membrane was incubated with blocking buffer for 30 min, chemiluminescent detection buffer, and antibody Anti-Digoxigenin-AP antibody solution diluted 1 : 10,000 solution for 30 min following the protocol of DIG Gel Shift Kit (3353591910, Roche, Basel, Switzerlan). The signals were visualized by luminescent image analyzer (ImageQuant LAS 4000 mini, Cytiva, Marlborough, MA, USA).

**Immunoprecipitation**. U2OS cells were transfected with pEGFP-C1, pEGFP-C1-FLAG-Ku70-WT, Ser77,78A, Ser77,78D, and Ser77,78E using Lipofectamine 2000 (Thermo Fisher Scientific, #11668019). Forty-eight hours after transfection, cells were irradiated with UV (200 J/m²) and cell extracts were obtained using lysis buffer (10 mM Tris-HCl pH 7.5, 100 mM NaCl, 0.5 mM EDTA, 0.5% NP-40, and 0.5% protease inhibitor cocktail). After 30 min, extracts were centrifuged at 20,000 × g for 10 min at 4 °C. Lysates were added to green fluorescent protein (GFP) trap beads_A (ChromoTek, gta-20, Munich, Germany). After 60 min of rotation at 4 °C, beads were washed with lysis buffer three times, added to an equal volume of sample buffer (0.1 M Tris-HCl pH 7.5, 4% SDS, 20% glycerol, 12% β-mercaptoethanol, and 1% bromophenol blue), and boiled at 95 °C for 10 min.

**Preparation of phosphorylated Ku70 antibody**. Phospho-Ser77/78 Ku70 peptide, KII[pS][pS]DRDL, to which a Cys residue was added at the C terminus was synthesized and conjugated to KLH (BEX, Tokyo, Japan). Rabbits were immunized one time by the phospho-Ser77/78 Ku70 peptide with Freund's Complete Adjuvant at the first week, and additionally three times by the peptide with Freund's Incomplete Adjuvant from the second to seventh weeks. Their serum was collected at 1 week after the final immunization and tested by ELISA using antigen phosphopeptides. Anti-phospho-Ser77/78 Ku70 antibody was purified by specific peptide column (order made by Cosmo Bio Co., Ltd, Tokyo, Japan).

**Isolation of human monoclonal anti-HMGB1-Abs**. The ADLib® system[49,50] was used to isolate HMGB1-specific human mAbs. The human ADLib® library (Chiome Bioscience, Inc., Tokyo, Japan), which was constructed by replacing the chicken IgM heavy and light chain loci in DT40 cells with the corresponding human IgG1 gene sequences, was screened for binding capability of membrane-bound IgG1 against biotinylated HMGB1 protein. Briefly, dsHMGB1 (HMGBiotech S.r.l., Milano, Italy) was biotinylated using the EZ-Link NHS-PEG4-Biotin kit (21455, Thermo Fisher Scientific, Waltham, MA, USA). This protein was then mixed with human ADLib® library cell suspension and incubated for 60 min at 4 °C. HMGB1-bound cells were enriched via magnetic cell isolation and stained with Alexa Fluor 647-conjugated streptavidin (1 : 1000, S21374, Thermo Fisher Scientific, Waltham, MA, USA), followed by flow cytometric single-cell sorting into 96-well plates containing Iscove's modified Dulbecco's medium supplemented with 9% fetal bovine serum (FBS) and 1% chicken serum (Thermo Fisher Scientific, Waltham, MA, USA) using a BD FACSAria™ Fusion system (BD Biosciences, San Jose, CA, USA). The plates were then incubated at 39.5 °C with 5% humidified CO₂ for 1 week to allow antibodies to accumulate in the media and the culture supernatants were later screened for positive anti-HMGB1-Ab via an ELISA. The sequences of the variable regions of the immunoglobulin genes from the isolated HMGB1-specific clones were determined from genomic DNAs. In some experiments, anti-HMGB1-Abs secreted into the culture supernatant from HMGB1-specific DT40 clones were purified via protein A chromatography (GE Healthcare, Chicago, IL, USA). Recombinant anti-HMGB1 IgG1 antibodies were transiently expressed by transfecting FreeStyle™ 293-F cells (Thermo Fisher Scientific, Waltham, MA, USA) with pFUSE-CHIg-hG1 (InvivoGen, San Diego, CA, USA) and pFUSE2-CLIg-hk (InvivoGen, San Diego, CA, USA) vectors carrying sequences encoding the variable regions of the heavy and light chains of HMGB1-specific clones, respectively, which were then purified via protein A chromatography.

**Epitope mapping**. Forty-two overlapping peptides that covered the complete sequence of human HMGB1 were synthesized and micro-plates (Maxisorp ELISA plates, Thermo Fisher Scientific, Waltham, MA, USA) were coated with 100 μg/mL of the peptides in PBS by overnight incubation at 4 °C. The plates were washed three times with PBS, blocked with 10 mg/mL bovine serum albumin (BSA) in PBS

(blocking buffer) overnight at 4 °C, and incubated with 30 μg/mL human monoclonal anti-HMGB1-Ab in blocking buffer overnight at 4 °C. After washing with PBS three times, the plates were incubated with horseradish peroxidase (HRP)-conjugated secondary antibody (anti-human IgG-HRP, #206, MBL, Nagoya, Japan) in blocking buffer (1 : 3000) for 2 h at 25 °C. The plates were washed five times with PBS and incubated with TMB solution (1-Step Ultra TMB-ELISA Substrate Solution, Thermo Fisher Scientific, Waltham, MA, USA) for 15 min. The reaction was stopped by 1× vol of stop solution (1 M H₂PO₃) and the optical density was measured at 450 nm.

**SPR analysis**. Biacore T100 (GE Healthcare, Chicago, IL, USA) was used for SPR analysis. For interaction between human dsHMGB1 and anti-human HMGB1-Ab, anti-human IgG (Fc) antibody (BR-1008-39, GE Healthcare, Chicago, IL, USA) was solved in in HBP-EP⁺ buffer and immobilized on a CM5 sensor chip until the amount of immobilized protein gained 6600 RU according to the standard protocol (GE Healthcare, Chicago, IL, USA); 1.5 μg/ml human anti-human HMGB1-Ab was injected for 120 s at a rate of 30 μl/min. Then, 0, 0.04, 0.2, 1, or 5 nM human dsHMGB1 (HM-120, HMGbiotech, Milano, Italy) was injected at the same speed for 90 s. To test the cross-reaction of human anti-human HMGB1-Ab to mouse dsHMGB1, mouse HMGB1 (#764004, Biolegend, San Diego, CA, USA) was pre-incubated with 50 μM of H₂O₂ at RT for 2 h, diluted with HBP-EP⁺ buffer, and injected to Biacore T100, as mouse dsHMGB1 was not commercially available.

For inhibition of the HMGB1-TLR4 or HMGB1-RAGE interaction by human monoclonal anti-HMGB1-Ab, human dsHMGB1 (50 μg/mL; 1.98 μM) were immobilized on a CM5 sensor chip to gain 5600 RU. After human monoclonal anti-HMGB1-Ab (50 μg/mL; 0.33 μM) was injected for 360 s at a rate of 30 μl/min, TLR4 (0–16 μg/ml) or RAGE (0–16 μg/ml) was injected for 90 s at a rate of 30 μl/min according to dual incubation protocol (GE Healthcare, Chicago, IL, USA). Regeneration of the sensor chip was performed by injection of 10 mM glycine-HCl pH 1.7 at the same flow rate for 60 s.

**ELISA-based calculation of $K_D$ value**. HMGB1 mAb (Chiome #129 antibody) was labeled with peroxidase (Peroxidase Labeling Kit-NH2, LK11, Dojindo, Kumamoto, Japan). For measurement, 1 nM dsHMGB1 was added to 96-well plates pre-coated with HMGB1-Ab (HMGB1 ELISA KIT II, Shino-Test, Tokyo, Japan) in PBS and incubated for 24 h at 37 °C. The plates were washed five times with wash buffer (HMGB1 ELISA KIT II, Shino-Test, Tokyo, Japan) and incubated for 2 h with peroxidase-conjugated human monoclonal anti-HMGB1-Ab (#129). After the plates were washed again, 100 μL of fluorescent agent (HMGB1 ELISA KIT II, Shino-Test, Tokyo, Japan) was added to wells and incubated for 30 min at room temperture. The reaction was stopped by stop solution (HMGB1 ELISA KIT II, Shino-Test, Tokyo, Japan) and the absorbance was measured by a plate reader (SPARK 10M, TECAN, Grodig, Austria) at 450 nm.

**Injection of human monoclonal anti-HMGB1-Ab**. 5xFAD or B6/SJL mice received subcutaneous injections of 0.1 μg/kg control IgG (human IgG isotype control, #12000C, Thermo Fisher Scientific, Waltham, MA, USA) or the human mono-clonal anti-HMGB1-Ab (Chiome #129 antibody) once a week, in the dorsal neck region from 6 to 8 months of age (nine times in total). In the case of intravenous injection, antibody #129 diluted in saline to a final volume of 50 μL was injected into the tail vein once a month from 6 to 8 months of age (three times in total).

**BBB transmittance of human monoclonal anti-HMGB1-Ab**. For detection of human monoclonal anti-HMGB1-Ab in the brain across BBB, we performed subcutaneous or intravenous injections of two different antibodies; non-labeled anti-HMGB1-Ab and biotinylated anti-HMGB1-Ab.

Non-labeled anti-HMGB1-Ab was detected in paraffin-embedded brain sections (5 μm thickness) that were obtained from 5xFAD and B6/SJL mice with or without the administration of human monoclonal anti-HMGB1-Ab (Chiome #129). After deparaffinization, rehydration, and antigen retrieval via boiled citrate buffer (0.1 M, pH 6.0), the sections were incubated in 10% normal goat serum in PBS for 1 h and then with biotin-conjugated anti-human IgG, labeled using the Biotin Labeling kit-NH2 (LK03, Dojindo, Kumamoto, Japan) for 3 h at 37 °C. For DAB reaction, the sections were further incubated with avidin-HRP (VECTASTAIN Elite ABC kit, #PK-6100, Vector, Burlingame, CA, USA) for 30 min at RT, followed by incubation with the DAB peroxidase substrate (DAB substrate kit, SK-4100, Vector, Burlingame, CA, USA) for 3 min at RT. For fluorescence detection, the sections were incubated with streptavidin-Alexa Fluor 488 (1 : 1000, S32354, Thermo Fisher Scientific, Waltham, MA USA) for 90 min at RT.

Biotinylated human monoclonal anti-HMGB1-Ab (labeled using the Biotin Labeling kit-NH2 (LK03, Dojindo, Kumamoto, Japan) was detected in paraffin-embedded brain sections (5 μm thickness) that were obtained from 5xFAD and B6/SJL mice with or without the administration of human monoclonal anti-HMGB1-Ab (Chiome #129). After deparaffinization and rehydration, the sections were incubated in 10% normal goat serum in PBS for 1 h and then with avidin-HRP (VECTASTAIN Elite ABC kit, #PK-6100, Vector, Burlingame, CA, USA) for 60 min at RT, followed by incubation with the DAB peroxidase substrate (DAB substrate kit, SK-4100, Vector, Burlingame, CA, USA) for 30 min at RT.

Images were acquired via light microscopy (Olympus BX53, Tokyo, Japan) or confocal microscopy (Olympus FV1200IX83, Tokyo, Japan).

**ELISA quantification of human monoclonal anti-HMGB1-Ab across BBB**. Non-labeled human monoclonal anti-HMGB1-Ab (#129) was quantified as below. Fifty microliters of rabbit anti-human IgG (#309-005-003, Jackson ImmunoResearch, PA, USA) at 1 μg/mL in PBS was added to each well of the plates (F96 MAXISORP NUNC-IMMUNO PLATE, #442404, Thermo Fisher Scientific, Waltham, MA, USA) and left overnight at 4 °C. After washing three times with PBS, 350 μL of PBS containing 3% BSA and 0.05% Tween-20 was added to each well followed by overnight incubation at 4 °C. The wells were then additionally washed with PBS containing 0.05% Tween-20. Brain tissue samples extracted in PBS containing 0.05% Tween-20 were added to the wells and reacted with the fixed anti-human IgG overnight at 4 °C. After the wells were washed three times with 0.05% Tween-20 in PBS, 50 μL of goat anti-human IgG-HRP (#206, MBL, Nagoya, Japan) in PBS containing 0.05% Tween-20 was added to each well followed by incubation for 2 h at RT. After the wells were washed three times with 0.05% Tween-20 in PBS and once with PBS, 50 μL of 1-Step Ultra TMB-ELISA (#34028, Thermo Fisher Scientific, Waltham, MA, USA) was added to start the reaction. After incubation for 30 min, the reaction was stopped via addition of 1 N HCl. The absorbance was measured using a plate reader (SPARK 10 M, TECAN, Grodig, Austria) at 450 nm. A standard curve for quantification was generated using multiple dilutions of antibody #129.

Biotinylated human monoclonal anti-HMGB1-Ab (#129) was quantified as below. Fixation of anti-human IgG, blocking, and reaction of samples were performed similarly. After the wells were washed three times with 0.05% Tween-20 in PBS and once with PBS, 50 μL of 1-Step Ultra TMB-ELISA was added to start the reaction. After 15 min, the reaction was stopped by addition of 1 N HCl. The absorbance was measured using a plate reader at 450 nm. A standard curve for quantification was generated using various concentration of biotinylated antibody #129.

**pSer46-MARCKS suppression in primary cortical neurons**. Briefly, cerebral cortexes from 15-day-old C57BL/6J mouse embryos were minced into fine pieces, rinsed with PBS, and incubated with 0.05% trypsin at 37 °C for 10–15 min and then with DNase at a final concentration of 25 μg/mL for another 5 min at 37 °C. The dissociated cells were washed twice with Dulbecco's modified Eagle's medium (DMEM; Gibco, NY, USA) containing 50% FBS, 25 mM D-glucose, 4 mM L-glutamine, and 25 mg/mL gentamycin. The cells were centrifuged at $100 \times g$ for 1 min, resuspended in 5 mL of 10% FBS/DMEM, gently triturated with blue tips, and filtered through a nylon mesh (Falcon 2350, BD Biosciences, NJ, USA). The isolated cells were centrifuged at $100 \times g$ again for 5 min and collected as a pellet. Finally, the cells were resuspended in Neurobasal medium and plated at $6 \times 10^4$ cells/well in eight-well chambered glass plates (Thermo Fisher Scientific, Waltham, MA, USA) coated with polyethyleneimine (Sigma-Aldrich, MO, USA). Neurons were incubated in 5% CO$_2$ at 37 °C. Twenty-four hours after plating, viruses were added at a multiplicity of infection of 5. Seventy-two hours after plating, arabinosyl cytosine (Sigma-Aldrich, St. Louis, MO, USA) was added to the culture medium (0.5 lM) to prevent unnecessary glial cell growth.

For the MARCKS-Ser46 phosphorylation inhibition assay, dsHMGB1 was used at 5 nM (HM-120, HMGbiotech, Milano, Italy) for 3 h. Antibody against HMGB1 was added at 25 nM 30 min prior to HMGB1 incubation. Three hours later, cells were collected and homogenized with a plastic homogenizer (Bio-Masher II, Nippi, Tokyo, Japan), followed by the addition of lysis buffer [100 mM Tris-HCl (pH 7.5, Sigma, MO, USA), 2% SDS (Sigma, St. Louis, MO, USA), 1 mM DTT (Sigma, St. Louis, MO, USA), and a protease inhibitor cocktail (Calbiochem, #539134, 1 : 200 dilution)]. The lysates were incubated on a rotator for 30 min at 4 °C and then boiled at 100 °C for 15 min. After centrifugation ($16,000 \times g \times 10$ min at 4 °C), the supernatants were diluted with an equal volume of sample buffer [125 mM Tris-HCl (pH 6.8, Sigma, St. Louis, MO, USA), 4% SDS (Sigma, St. Louis, MO, USA), 20% glycerol (Wako, Osaka, Japan), 12% mercaptoethanol (Wako, Osaka, Japan), and 0.05% BPB (Nacalai, Kyoto, Japan)]. SDS-PAGE and western blot analysis was then performed as described below.

**Western blot analysis**. Mouse cerebral cortex tissues or primary cortical neurons were homogenized on ice using a Dounce glass homogenizer (1 mL tissue grinder, #357538, Wheaton, NJ, USA) in lysis buffer [100 mM Tris-HCl (pH 7.5, Sigma, St. Louis, MO, USA), 2% SDS (Sigma, St. Louis, MO, USA), 1 mM DTT (Sigma, St. Louis, MO, USA), and a protease inhibitor cocktail (Calbiochem, #539134, 1 : 200 dilution)]. The lysates were incubated on a rotator for 30 min at 4 °C and then boiled at 100 °C for 15 min. After centrifugation ($16,000 \times g \times 10$ min at 4 °C), the supernatants were diluted with an equal volume of sample buffer [125 mM Tris-HCl (pH 6.8, Sigma, St. Louis, MO, USA), 4% SDS (Sigma, MO, USA), 20% glycerol (Wako, Osaka, Japan), 12% mercaptoethanol (Wako, Osaka, Japan), and 0.05% BPB (Nacalai, Kyoto, Japan)]. The samples were separated by SDS-PAGE, transferred to Immobilon-P polyvinylidene difluoride membranes (Millipore, Burlington, MA, USA) using a semi-dry method, and then blocked with 2% BSA (Nacalai, Kyoto, Japan) or 5% milk in TBST (10 mM Tris-HCl (pH 8.0,Sigma, St. Louis, MO, USA), 150 mM NaCl, 0.05% Tween-20). The following primary and secondary antibodies were diluted in TBST with 0.2% BSA or in Can Get Signal solution (Toyobo, Osaka, Japan): rabbit anti-phospho-Ku70 (Ser77/Ser78) (1 : 100,000 [ordered from Cosmo Bio Co., Ltd, Tokyo, Japan]); rabbit anti-GST(z-5) (1 : 5000, sc-459, Santa Cruz Biotechnology, Dallas, TX, USA); mouse anti-amyloid β (1 : 5000, clone 82E1, #10323, IBL, Gumma, Japan); mouse anti-Histone H4 (1 : 5000, ab31830, Abcam, Cambridge, UK); mouse anti-β-actin (1 : 1000, sc-8334, Santa Cruz Biotechnology, TX, USA); rabbit anti-phosphorylated Ser46-MARCKS (1 : 100,000, GL Biochem Ltd, Shanghai, China); mouse anti-MARCKS (1 : 1000, sc-100777, Santa Cruz Biotechnology, TX, USA); mouse anti-amyloid β (1 : 3000, clone 82E1, IBL, Gumma, Japan); mouse anti-phospho-H2AX (γH2AX) (1 : 3000, JBW301, Millipore, Burlington, MA, USA); rabbit anti-53BP1 (1 : 15,000, NB100-304, Novus, CO, USA); rabbit anti-phospho-p44/42 MAPK (Erk1/2) (phospho Thr202/Tyr204) (1 : 30,000, 4370, Cell Signaling Technology, Danvers, MA, USA); rabbit anti-p44/42 MAPK (Erk1/2) (1 : 5000, 9102, Cell Signaling, Danvers, MA, USA); rabbit anti-phospho-CDK1 (phospho T14) (1 : 5000, ab58509, Abcam, Cambridge, UK); mouse anti-CDK1 (1 : 5000, ab18am, Abcam); rabbit anti-Ku70(phosphor Ser5) (1 : 3000, PA5-40427, Thermo Fisher Scientific, Waltham, MA, USA); goat anti-Ku70 (M-19) (1 : 3000, sc-1487, Santa Cruz Biotechnology, Dallas, TX, USA); mouse anti-phospho-ATM (phospho S1981) (1 : 10,000, 200-301-400, Rockland, PA, USA); rabbit anti-ATM (1 : 3000, PC85-100UG, EMD Chemicals, MA, USA); rabbit anti-phospho-ATR (phospho Ser428) (1 : 5000, 2853, Cell Signaling, Danvers, MA, USA); mouse anti-ATR (C-1) (1 : 3000, sc-515173, Santa Cruz Biotechnology, Dallas, TX, USA); rabbit anti-phospho-Tau (phospho Ser214) (1 : 15,000, ab170892, Abcam, Cambridge, UK); rabbit anti-phospho-PKCα (phospho Thr638) (1 : 20,000 ab32502, Abcam, Cambridge, UK); rabbit anti-phospho-PKCβI (phospho Thr641) (1 : 1000, sc-101776, Santa Cruz Biotechnology, Dallas, TX, USA); goat anti-phospho-PKCβII/δ (phospho Thr660) (1 : 10,000, sc-11760, Santa Cruz Biotechnology, Dallas, TX, USA); rabbit anti-phospho-MEK1/2 (phospho Ser217/221) (1 : 50,000, 9121, Cell Signaling Technology, Dallas, TX, USA); rabbit-MEK1 (1 : 3000, ab32091, Abcam, Dallas, TX, USA); rabbit anti-pan-PKC (1 : 1500, GTX52352, GeneTex, Irvine, CA, USA); rabbit anti-Ku70(acetyl K331) antibody (1 : 5000, ab190626, Abcam, Cambridge, UK); mouse anti SIRT1 (1 : 2000, #8469 S, Cell Signaling Technology, Danvers, MA, USA); rabbit anti-14-3-3 (1 : 1000, #14503-1-AP, Protein Tech); rabbit anti-Ku80 (1 : 1000, sc-9034, Santa Cruz Biotechnology, Dallas, TX, USA); rabbit anti-DNA PKcs (1 : 5000, ab32566, Abcam, Cambridge, UK); rabbit anti-TDP43 (1 : 1000, ab109535, Abcam, Cambridge, UK); VCP (1 : 1000, #612182, BD Bioscience, San Jose, CA, USA); rabbit anti-enhanced GFP (EGFP) (1 : 5000, sc-8334, Santa Cruz Biotechnology, Dallas, TX, USA); mouse anti-Tau (1 : 5000, ab80579, Abcam, Cambridge, UK); HRP-linked anti-rabbit IgG (1 : 3000, NA934, GE Healthcare, Buckinghamshire, UK); HRP-linked anti-mouse IgG (1 : 3000, NA931, GE Healthcare, Buckinghamshire, UK); and donkey anti-goat IgG-HRP (1 : 3000, sc-2020, Santa Cruz Biotechnology, Dallas, TX, USA) antibodies. Membranes were incubated with primary and secondary antibodies overnight at 4 °C and for 1 h at RT, respectively. The ECL Prime Western Blotting Detection Reagent (RPN2232, GE Healthcare, Chicago, IL, USA) and a luminescent image analyzer (ImageQuant LAS 500, GE Healthcare, Chicago, IL, USA) were used to detect proteins. Uncropped western blotting images were included in Supplementary Fig. 15.

**Immunohistochemistry**. For the immunohistochemistry experiments, mouse and human brain tissue samples were fixed with 4% paraformaldehyde and embedded in paraffin. Sagittal or coronal sections (5 μm thickness) were obtained using a microtome (Yamato Kohki Industrial Co., Ltd, Saitama, Japan). Immunohistochemistry was performed using the following primary antibodies: rabbit anti-phosphorylated Ser46-MARCKS (1 : 2000 [ordered from GL Biochem (Shanghai) Ltd, Shanghai, China]); mouse anti-amyloid β (1 : 1000, clone 82E1, #10323, IBL, Gumma, Japan); rabbit anti-phospho-Ku70 (Ser77/Ser78) (1 : 1000 [ordered from Cosmo Bio Co., Ltd, Tokyo, Japan]); mouse anti-Ku70 (1 : 250, E-5, Santa Cruz, Dallas, TX, USA); mouse anti-MAP2 (1 : 200, sc-32791, Santa Cruz Biotechnology, TX, USA); rabbit anti-MAP2 (1 : 2000, ab32454, Abcam, Cambridge, UK); mouse anti-phospho-H2AX (γH2AX) (1 : 300, JBW301, Millipore, MA, USA); rabbit anti-53BP1 antibody (1 : 5000, NB100-304, Novus, CO, USA); rabbit anti-YAP (1 : 50, GTX129151, GeneTex, Irvine, CA, USA); rabbit anti-NFkB (1 : 100, #8242, Cell Signaling Technology, Danvers, MA, USA); mouse anti IL-6 (1 : 50, ab208113, Abcam, Cambridge, UK); rabbit anti-CCL2/MCP1 (1 : 100, NBP1-07035, Novus, CO, USA); rabbit anti-pSer232-RIP3 (1 : 200, ab195117, Abcam, Cambridge, UK); rabbit anti-pSer345-MLKL (1 : 500, ab196436, Abcam, Cambridge, UK); rabbit anti-Lamin B1 antibody (1 : 5000, ab16048, Abcam, Cambridge, UK); rabbit anti-cleaved Caspase1 (1 : 100, #89332, Cell Signaling Technology, Danvers, MA, USA); rabbit anti-cleaved Caspase9 (1 : 200, #9509, Cell Signaling Technology, Danvers, MA, USA); goat anti-Iba1 (1 : 500, #011-27991, Wako, Osaka, Japan); rabbit anti-Iba1 (1 : 100, #019-19741, Wako, Osaka, Japan); mouse anti-GFAP Cy3-conjugated (1 : 5000, #C9205, Sigma-Aldrich, St. Louis, MO, USA); mouse anti-NeuN (1 : 1000, ab104224, Abcam, Cambridge, UK); rabbit anti-PSD95 (1 : 100, D74D3, Cell Signaling Technology, Danvers, MA, USA); and rabbit anti-phospho-Tau (phospho S214) (1 : 2000, ab170892, Abcam, Cambridge, UK) antibodies. The reaction products were visualized by Alexa Fluor 488-, 568-, and 647- conjugated secondary antibody (1 : 1000, Molecular Probes, MA, USA). Nuclei were stained with 4′,6-diamidino-2-phenylindole (0.2 μg/mL in PBS, #D523, DOJINDO Laboratories, Kumamoto, Japan). All images were acquired using confocal microscopy (Olympus

FV1200IX83, Tokyo, Japan). In the case of HE staining, images were obtained using light microscopy (Olympus BX53 with DP72 digital camera, Tokyo, Japan).

**Y-maze test**. Exploratory behavior was performed in a Y-shaped maze consisting of three identical arms with equal angles between each arm (O'HARA & Co., Ltd, Tokyo, Japan). Outer Radius is 41 cm and each arm is separated by 120° angles. In each arm position, luminous intensity is adjusted to 15 lux on average. We performed at dark phase (20:00–8:00). In order to chase the mice movement in this condition, we adjusted the luminance between 70 and 180 steps in 8-bit image (0–255 step gray scale). Before starting the test, we kept mice in the Y-maze test room where the brightness was adjusted to the same lux (15 lux on average) for 1 h, to allow them to acclimatize to the circumstances. Eight-month-old mice were placed at the end of one arm and allowed to move freely through the maze during an 8 min session. The percentage of spontaneous alterations (reported as the alteration rate) was calculated by dividing the number of entries into a new arm that was different from the previous one by the total number of transfers from one arm to another arm.

**In vitro phosphorylation**. Next, 10 pmol of GST-human Ku70 (Abnova, #H00002547-P01, Taipei City, Taiwan) was incubated with each kinase (PKCα, PKCβI, PKCβII, PKCγ, MEK1, and ERK1) for 1 h at 30 °C in 50 μl reaction buffer (5 mM MOPS pH 7.2, 2.5 mM β-glycero-phosphate, 5 mM MgCl₂, 1 mM EGTA, 0.04 mM EDTA, 0.05 mM DTT, and 50 μM ATP). For PKCs, 50 μg/ml phosphatidylserine (P7769, Sigma-Aldrich, St. Louis, MO, USA), 5 μg/ml 1-Oleoyl-2-acetyl-sn-glycerol (O6754, Sigma-Aldrich, St. Louis, MO, USA), and 0.1 mM CaCl₂ were further added to the reaction buffer. For CDK1, 10 pmol of GST-Ku70 was incubated in 50 μl of different reaction buffer (5 mM HEPES pH 7.0, 0.5 mM MgCl₂, 1 mM MgCl₂, 0.1 mM DTT, and 50 μM ATP). After reaction, samples were mixed with 1× vol. of sample buffer and directly subjected to SDS-PAGE.

**Two-photon microscopy**. Adeno-associated virus 9-AcGFP carrying the synapsin I promoter (titer: $1 \times 10^{10}$ vector genomes/mL, 1 μL) and AAV2-VAMP2-mCherry carrying the cytomegalovirus promoter were injected into two neighboring positions in the retrosplenial cortex −1.0 mm from the bregma (mediolateral, 0.5 mm; depth, 1 mm) and −3.0 mm from the bregma (mediolateral, 0.5 mm; depth, 1 mm), respectively, under anesthesia with 1% isoflurane. In the rescue experiment, 5xFAD, B6/SJL, APP-KI ($App^{NL-G-F/NL-G-F}$), or B6 mice received subcutaneous injections of 1 μg/kg control IgG (human IgG isotype control) or the human monoclonal anti-HMGB1-Ab Chiome #129 antibody), according to the protocol described in Supplementary Fig. 4. After the antibody injection, the skull on retrosplenial cortex was thinned using a high-speed micro-drill. The head of each mouse adhered to the head plate was immobilized by a stage mounted on the microscope table. Two-photon imaging was performed using a laser-scanning microscope system FV1000MPE2 (Olympus, Tokyo, Japan) equipped with an upright microscope (BX61WI, Olympus, Tokyo, Japan), a water-immersion objective lens (XLPlanN25xW; numerical aperture, 1.05), and a pulsed laser (MaiTaiHP DeepSee, Spectra Physics, Santa Clara, CA, USA)[40,64]. EGFP and mCherry were excited at 920 nm and scanned at 495–540 nm and 575–630 nm, respectively. High-magnification imaging (101.28 × 101.28 μm; 1024 × 1024 pixels; 1 μm Z step) of cortical layer I was performed with a 5× digital zoom via the thinned-skull window in the retrosplenial cortex[100].

**Time-lapse Ku70 imaging**. Laser microirradiation and signal acquisition from the resulting damage sites were performed according to a previously described method[101,102]. U2OS cells were seeded on 35 mm glass-bottomed dishes (D11140H, Matsunami Glass IND., Ltd, Osaka, Japan) containing 2 mL of culture medium (DMEM, D5796, Sigma-Aldrich, St. Louis, MO, USA). On the next day, 2.5 μg of EGFP-C1-FLAG-Ku70 plasmid (#46957, Addgene, Watertown, MA, USA) was transfected to cells by Lipofectamine LTX (Thermo Fisher Scientific, Waltham, MA, USA). For the siRNA treatment, the medium was changed 24 h after plasmid transfection and 15 nM (208 ng/mL) siRNA was transfected to cells using Lipofectamine RNAiMAX (Thermo Fisher Scientific, Waltham, MA, USA). For knockdown of human TLR4, RAGE, and MEK1, custom-made siRNAs of the following sequences were generated by Nihon Gene Research Laboratories (Miyagi, Japan): (human TLR4) sense: 5′-GGUGUAUCUUUGAAUAUGAGTT-3′, antisense: 5′-CUCAUAUUCAAAGAUACACCTT-3′; (human RAGE) sense: 5′-CGG CUGGUGUUCCAAUAAUT-3′, antisense: 5′-UUAUUGGAACACCAGCCGTG-3′; (human MEK1) sense: 5′-GCAACUCAUGGUUCAUGCUTT-3′, antisense: 5′-AGCAUGAACCAUGAGUUGCTT-3′. The siRNAs targeting human ERK1 and ERK2 were purchased from Origene (ERK1: SR303752; ERK2: SR303751; MD, USA). Forty-eight hours after EGFP-Ku70 transfection, dsHMGB1 (6 ng/mL, 0.24 nM) and/or anti-HMGB1-Ab (Chiome #129, 6 ng/mL, 0.04 nM) were added to the medium 3 h before irradiation and time-lapse imaging. Gö6976 (10 or 100 nM) or anti-HMGB1-Ab (Chiome #129, 6 ng/mL, 40 nM) were incubated for 3.5 h before irradiation and time-lapse imaging. Cells were treated with 2 μM Hoechst33258 (DOJINDO, Kumamoto, Japan) for 20 min to sensitize the cells to DSBs. Laser microirradiation was performed with a confocal microscope (FV1200IX83, Olympus, Tokyo, Japan) under live-cell imaging conditions (37 °C, 5% CO₂) maintained in a stage-top incubator (Tokai Hit, Shizuoka, Japan) and the

rectangle-shaped areas over cell nuclei were irradiated using a 405 nm laser diode (40× objective lens, 95%, 100 scans, 12.5 units/pixel). Time-lapse images were obtained every 30 s for 10 min.

**Differentiation of iPS cell-derived pan-neuron**. Human iPS cells carrying *APP* KM670/671NL mutations were generated by genome editing as described previously[64]. Normal human iPS (ASE-9203, Applied StemCell, Inc., CA, USA) and *APP* KM670/671NL iPS cells were cultured in TeSR-E8 medium (STEMCELL Technologies, BC, Canada) with 10 μM Y27632 (253-00513, Wako, Osaka, Japan). On the next day, medium was changed to Stem Fit medium (AK02N, Ajinomoto, Tokyo, Japan) with 3 μM SB431542 (13031, Cayman Chemical, Ann Arbor, MI, USA), 3 μM CHIR99021(13122, Cayman Chemical, Ann Arbor, MI, USA), and 3 μM dorsomorphin (044-33751, Wako, Osaka, Japan). After another 6 days, the iPS cells were dissociated to single cells using TrypLE Select (12563-011, Thermo Fisher Scientific, MA, USA) with 10 μM Y27632. Resultant neurospheres were cultured in KBM medium (16050100, KHOJIN BIO, Saitama, Japan) with 20 ng/ mL Human-FGF-basic (100-18B, Peprotech, London, UK), 10 ng/mL Recombinant Human LIF (NU0013-1, Nacalai, Kyoto, Japan), 10 μM Y27632 (253-00513, Wako, Osaka, Japan), 3 μM CHIR99021 (13122, Cayman Chemical, Ann Arbor, MI, USA), and 2 μM SB431542 (13031, Cayman Chemical, Ann Arbor, MI, USA), and passaged twice. Then, neurospheres were dissociated into single cells, seeded on eight-well chambers coated with poly-L-ornithine (P3655, Sigma-Aldrich, St. Louis, MO, USA) and laminin (23016015, Thermo Fisher Scientific, Waltham, MA, USA), and cultured in DMEM/F12 (D6421, Sigma-Aldrich, St. Louis, MO, USA) supplemented with B27 (17504044, Thermo Fisher Scientific, Waltham, MA, USA), Glutamax (35050061, Thermo Fisher Scientific, Waltham, MA, USA), and penicillin/streptomycin (15140-122, Thermo Fisher Scientific, Waltham, MA, USA) for 5 days.

**RNA sequencing**. Total RNA samples were extracted from normal or $APP^{KM670/ 671NL}$ iPSC-derived neurons using NucleoSpin® RNA (740955.10, MACHEREY-NAGEL, Düren, Germany). RNA sequencing was performed by Rhelixa (Tokyo, Japan). Trimmomatic (version 0.39-1) with options "-PE -phred33 ILLUMINA-CLIP:TrueSeq3-PE.fa:2:30:10" was used to remove low quality or "N" bases.

Hisat2 2.2.1 was used to map short reads to human reference genome (Hg38).

**Immunocytochemistry of cilia**. iPSC-derived neurons were fixed in 4% formaldehyde and then permeabilized by incubation with 0.1% Triton X-100 in PBS for 6 min at RT. After blocking with blocking buffer (2% donkey serum and 0.1% Triton X-100) containing 10 mg/mL BSA for 30 min at RT, sections were incubated with primary antibody for 16 h at 4 °C and with secondary antibodies for 60 min at RT. The antibodies used for immunocytochemistry were as follows: rabbit anti-AC3 (1 : 500, sc-588, Santa Cruz Biotechnology, Dallas, TX, USA), mouse anti-MAP2 (1 : 250, sc-32791, Santa Cruz Biotechnology, Dallas, TX, USA), Alexa Fluor 488-conjugated anti-rabbit IgG (1 : 1000, A21206, Molecular Probes, Eugene, OR, USA), and Alexa Fluor 647-conjugated anti-mouse IgG (1 : 1000, A31571, Molecular Probes, Eugene, OR, USA) antibodies.

**Generation of PPI-based networks connecting phosphorylated proteins**. To generate PPI-based networks, changes in the levels of phosphopeptides at each time point were compared between AD and control mice using Welch's test with the Benjamini–Hochberg post hoc procedure. Statistical significance was set at $q < 0.05$ and phosphopeptides with significant changes were used to identify the corresponding proteins. UniProt Accession numbers were added to the proteins with altered phosphorylation patterns. Proteins whose UniProt IDs were not listed in the Human Genome Project (Genome network project: GNP) (http://genomenetwork.nig.ac.jp/ index_e.html) databases were removed from the list of altered proteins. The selected proteins were then used to generate the PPI-based network of AD at each time point based on the integrated GNP databases, including BIND (http://www.bind.ca/), BioGrid (http://www.thebiogrid.org/), HPRD (http://www.hprd.org/), IntAct (http:// www.ebi.ac.uk/intact/site/index.jsf), and MINT (http://mint.bio.uniroma2.it/mint/ Welcome.do). A database of GNP-collected information was created on the supercomputer system available at the Human Genome Center of the University of Tokyo. The PPI network was visualized using Cell Illustrator[103] and the representative quantity (geometric mean of the peptide ratios of significantly changed phosphopeptides) and lowest q-value were attached to each node.

**Centrality analysis of the PPI-based network**. The betweenness centrality was used to select core nodes from the PPI-based network of altered phosphoproteins. The betweenness centrality is defined as the ratio of the node appearing in all the shortest paths between any two nodes in the network. A node with high betweenness centrality can be interpreted as one that is connected with many nodes in terms of the information flow of the paths.

To calculate the betweenness centrality score of the nodes in the PPI-based network of altered phosphoproteins in AD over the various time points, the lists of altered proteins at each time point were merged and a PPI-based AD network was generated using the merged list. The calculated betweenness centrality scores were

converted to z-scores to evaluate the importance of the proteins in the generated network.

**Statistics and reproducibility**. For the phosphoproteome analysis, the peptide ratios followed a log-normal distribution. Therefore, the data are presented as the geometric mean ± the SEM and the differences between the AD and control samples were tested using the log ratio test. A two-tailed Welch's test with the Benjamini–Hochberg post hoc procedure was applied to compare the differences in changes of phosphopeptides between AD and control samples. The significance level was set at 5%. For biological experiments, the data were assumed to follow a normal distribution and are presented as the mean ± SEM. Student's t-test was applied for two-group comparisons. For multiple-group comparisons, Tukey's honestly significant difference (HSD) test or Dunnett's comparison was applied. The significance level was set at 1% or 5%. Sample size, which was determined from previous reports from our and others' laboratories, is described in each figure and figure legend.

**Ethics for animal experiments**. This study was performed in strict accordance with the ARRIVE guidelines (Animal Research: Reporting in vivo Experiments) for the Care and Use of Laboratory Animals of the National Institutes of Health. It was approved by the Committees on Gene Recombination Experiments and Animal Experiments of Tokyo Medical and Dental University (G2018-082C and A2019-218C2).

**Ethics for human experiments**. All experiments with human samples were performed after obtaining informed consent and were carried out in accordance with the approved guidelines for human experimental research. The experiments were approved by the Committee on Human Ethics of the Tokyo Medical and Dental University (O2014-005-13/O2020-002, O2017-008-02).

**Reporting summary**. Further information on research design is available in the Nature Research Reporting Summary linked to this article.

## Data availability

The mass spectrometry proteomics data of 5xFAD mice have been deposited to the ProteomeXchange Consortium via the PRIDE partner repository with the data set identifier PXD001292[40]. Full-size figures of Supplementary Fig. 5b are available in the author's website (http://suppl.atgc.info/031/). The mass spectrometry proteomics data of HMGB1-treated U2OS cells and human postmortem brains have been deposited to the ProteomeXchange Consortium via the PRIDE partner repository with the data set identifier PXD028089. Plasmids will be delivered under MTA when requested. All source data including RNA sequencing are shown in Supplementary Data 1.

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

## Acknowledgements

We thank technical support of Huang Yong for technical assistance. This work was supported by Grant-in-Aid for Scientific Research from Japan Society for Promotion of Science (JSPS) (16H02655; 19H01042), a Grant-in-Aid for Scientific Research on Innovative Areas (Foundation of Synapse and Neurocircuit Pathology, 22110001/ 22110002) from the Ministry of Education, Culture, Sports, Science and Technology of Japan (MEXT), and Brain Mapping by Integrated Neurotechnologies for Disease Studies (Brain/MINDS) from the Japan Agency for Medical Research and Development (AMED) (JP18dm0207013h0005) to H.O. Human brain samples were obtained from Brain Bank for Aging Research in Tokyo Metropolitan Geriatric Hospital and Institute of Gerontology which is supported by JSPS KAKENHI Grant Number JP 16H06277 (CoBiA) and AMED under Grant Number JP21wm0425019. MD simulation was partially supported by Basis for Supporting Innovative Drug Discovery and Life Science Research (BINDS) from AMED (3018).

## Author contributions

H.T. and K.F. performed experiments, analyzed data, and wrote the paper. K.K., K.T., X.J., M.J., Y.Y. and S.T. performed experiments and analyzed data. H.M., R.T. and Y.N. screened HMGB1 antibody as CRO. S.M., T. Saito, and T. Saido prepared human or mouse samples. T.I. and N.I. supported experiments of structural biology. Y.Y. and K.T. supported MD simulation. M.B. advised HMGB1 research. H.H. designed and performed mathematical experiments, and wrote the paper. H.O. designed this project, co-designed experiments, analyzed data, and wrote the paper.

## Competing interests

H.M., R.T. and Y.N. are employees of Chiome Co. Ltd. The remaining authors declare no competing interests.
