## [Peer Review File · Communications Biology]

Reviewers' comments:

Reviewer #1 (Remarks to the Author):

This is an interesting and novel paper that expands the knowledge of DNA damage and the induction of secondary apoptosis in Alzheimer's disease (AD), and expands on the complex role of HMGB1 in these processes. Their work shows that abnormal phosphorylation of Ku70 at Ser77/78 alters its DNA binding dynamics and prevents the recruitment of Ku70 to double-stranded breaks in the DNA. This phosphorylation can be driven through an extracellular HMGB1/TLR4/PKC α axis. This work elucidates a drug and antibody targetable process that drives and propagates DNA damage from necrotic and hyper-activated neurons to bystander cells, can impair neuronal cilia and promote neuronal dysfunction, and promoting clusters of neuronal death in AD. It provides a convincing mechanism for bystander cell death that explains the clusters of neuronal death seen in AD. The use of drug and antibody intervention in this process, even in post-symptomatic mice, may have clinical relevance in the future.

The contents of the paper are important to scientists working on AD and DNA damage, and has secondary interest for those working on the role of HMGB1 in the damage-associated molecular patterns (DAMP) and senescence fields.

The data is clear and appears technically sound, with the experiments well-reasoned and controlled. I don't have the expertise to comment on the technical aspects specifically relating to phosphorylation and generation of the antibody in extended figure 1 and 2, however, so would bow to any comments from other reviewers on those experiments. I have no other major comments on contents of the experiments except for one point on the Y-maze. There are some other minor points that should be addressed, listed below.

1. Minor grammatical errors in introduction and discussion. It would benefit from some proof reading to aid clarity in some places, as there are a few awkward and long sentences.
2. Please make sure acronyms are expanded on first use, e.g. transcriptional repression-induced atypical cell death of neuron (TRIAD).
3. In the introduction: "DNA damage accumulates also in senescent cells, which secrete a set of chemokines and cytokines collectively known as senescence-associated secretory phenotype (SASP)^{21–24}"
 - a. The authors may consider citing a recent review on the SASP here for simplicity, ideally one that addresses SASP in stress-induced (in addition to oncogenic) senescence.
4. Given the mention of senescence and the focus of the paper, a brief sentence on the presence of senescent glial and neuronal cells in neurodegeneration and the ageing brain should be included. See works from labs of Miranda Orr, Diana Jurk, Julie Andersen, etc. I have some more comments on this in points 8-10.
5. Figure 6 B/C seem to be mixed up in the text. I would suggest double checking if that's correct.
6. Some of the text in the figures is quite small due to the amount of data and may be difficult for readers with visual impairments. The current resolution is sufficient that you can zoom in on them, however.
7. Methods for Y-Maze should really state dimensions of the maze, phase of light-cycle the experiment was performed in (and preferably light intensity). For those from a behavioural background looking at the data, handling technique (tail vs tube handling) and whether mice were allowed to acclimatise to the experimental room are also useful to know.
 - a. If it was performed during the light cycle, stress from handling, high light intensities and waking them during their sleep cycle can lead to increased freezing, defecation and time spent in the 'home arm' (i.e. the arm into which the mouse was initially placed). This can influence the

assessment of working memory, and can drive alteration ratios below 50% (due to shuttling from the 'home arm' to one other arm, without exploration of the third arm). I would suggest the authors consider performing trialling the use of the Y-Maze during the dark phase if they have not already tried it (under dim or red-light, <2 lux), as they should get cleaner data on memory. In my own hands, control mice of this age (C57Bl/6) perform much better in this set up (80-100% alternation) in a 40cm long arm maze and 8 minute trial, while impaired mice still show poor performance.

b. I would not consider this a barrier to publication (rather intended as a helpful suggestion for future work), as performing Y-maze during the light cycle is still common practice in the field. The data they show still looks convincing, but they could mention that there may some confounding influences from anxiety. It would also be entirely unreasonable to ask them to repeat this part of the study

c. A dotted line through the 50% mark on the y-axis would also be preferred, to show expected result from random alternation (i.e. from dysfunctional memory).

8. While the authors mention cellular senescence, the senescence associated secretory phenotype (SASP) and its role in the release of HMGB1 to the extracellular space in the introduction, they do not address it later in the paper. HMGB1 has been identified as important component of some senescent and SASP phenotypes, and this work may show how this could drive bystander DNA damage from senescent cells in some populations (previously largely assumed to be due to reactive species production by senescent cells). Especially given the increasing evidence of the role of senescence in glial and neuronal populations driving early pathology in AD and cognitive decline, the authors could discuss this in the discussion.

9. It would be nice to have some evidence to address this in the paper. A relatively limited evaluation of senescent and inflammatory markers, and discussion would suffice if the authors wish to pursue this - any more in-depth investigation would be enough work to warrant another paper. It would be interesting to see if treatment with anti-HGMB1 antibody or Gö6976 alters levels of senescence markers (as might be expected, with interference with paracrine signalling and reduced DNA damage). If they choose this, I suggest sticking to a few relatively simple markers:

- p16 and p21 - can be determined in homogenate, or by RNA fluorescence in situ.
- Lamin B1 - can be performed in immunohistochemistry (positive/negative or intensity)
- Nuclear size - can be taken from existing images.
- γH2A.X - data is already present.
- I would not suggest Sen-β-Gal in neuronal cells. HMGB1 sub-cellular localisation by immunohistochemistry (nuclear exclusion) is also sometimes used, but as this paper shows it has many roles.

10. Assessment of inflammatory markers (senescence associated or otherwise) would also be nice to have, as reductions through interruption of this cascade and inflammatory and SASP-signalling may contribute towards the improved phenotype. Assessment of glial cells would also be nice, as HMGB1 released by neurons (amongst other cells) can drive chronic inflammation and neurodegeneration. Markers of microglia (e.g. iba1) and microglial activation could be considered, as well as astrocytes if time permits. Reductions in inflammation from reduced extracellular HMGB1 signalling may be partially behind the improvements seen in treated mice (I suspect it will actually be an interplay between all these factors in forming/breaking a vicious cycle of destruction).

11. I would be interested in the author's speculation on why this mechanism exists, whether it has a physiological role or is purely aberrant (i.e. why have a DAMP signal that reduces DNA repair in recipient cells, perhaps useful in countering parasitic and viral infection?). This may be a bit speculative for the discussion in this paper, I am simply interested in their thoughts.

Reviewer #2 (Remarks to the Author):

In this manuscript, Tanaka et al studied the mechanism of the increase in DNA damage in Alzheimer's Disease (AD). They used comprehensive phosphoproteome analysis and other techniques to study abnormal phosphorylation of Ku70 at Ser77/78 and its effects in Ku70-DNA interaction using human AD postmortem brains. While I am not familiar of this field and the novelty of this particular study with respect to the existing literature, it is evident to me that this is a carefully executed research. The experiments are planned well and the results are nicely explained. The following are some of my concerns, which I request the authors to consider:

a) One of the important inference of this paper has come from the simulations using Pymol. Pymol offers only limited scope for any serious simulations, which needs to be complimented with a MD (or similar) simulations using a reasonable time scale.

b) Some of the figures, which contain protein structures (for example, Extended data 1), are difficult to follow. Since these figures are important for this manuscript, the authors should find a way to increase the size and/or the resolutions.

c) SPR experiments are not carefully analyzed. For example, there is a number ($1.8 \times 10^{-8} \text{ M}$) provided for the data of #213-012 (Extended data 2D), I am not sure from the figure, if these numbers are reliable. The authors should provide some estimate of their fitting reliability, like residual distributions etc. The authors have fitted the linear inhibition data (Extended data 2H, the red curve for RAGE) with a non-linear equation, which is not correct. The authors should increase the dose and achieve saturation, or they should remove these numbers.

d) How many times the SPR experiments were carried out? Not a single table has any error these measurements. Neither there is any error bars in the figures for these measurements.

e) Since the binding parameters are determined mostly from SPR, it would be nice if some of these data could be complemented using another assay.

Reviewers' comments:

Reviewer #1 (Remarks to the Author):

This is an interesting and novel paper that expands the knowledge of DNA damage and the induction of secondary apoptosis in Alzheimer's disease (AD), and expands on the complex role of HMGB1 in these processes. Their work shows that abnormal phosphorylation of Ku70 at Ser77/78 alters its DNA binding dynamics and prevents the recruitment of Ku70 to double-stranded breaks in the DNA. This phosphorylation can be driven through an extracellular HMGB1/TLR4/PKC α axis. This work elucidates a drug and antibody targetable process that drives and propagates DNA damage from necrotic and hyper-activated neurons to bystander cells, can impair neuronal cilia and promote neuronal dysfunction, and promoting clusters of neuronal death in AD. It provides a convincing mechanism for bystander cell death that explains the clusters of neuronal death seen in AD. The use of drug and antibody intervention in this process, even in post-symptomatic mice, may have clinical relevance in the future.

The contents of the paper are important to scientists working on AD and DNA damage, and has secondary interest for those working on the role of HMGB1 in the damage-associated molecular patterns (DAMP) and senescence fields.

The data is clear and appears technically sound, with the experiments well-reasoned and controlled. I don't have the expertise to comment on the technical aspects specifically relating to phosphorylation and generation of the antibody in extended figure 1 and 2, however, so would bow to any comments from other reviewers on those experiments. I have no other major comments on

contents of the experiments except for one point on the Y-maze. There are some other minor points that should be addressed, listed below.

>>> Thank you very much for your exact understanding and kind evaluation of our paper.

1. Minor grammatical errors in introduction and discussion. It would benefit from some proof reading to aid clarity in some places, as there are a few awkward and long sentences.

>>> A professional native editor reviewed the manuscript. We hope such grammatical errors were corrected in introduction and discussion.

2. Please make sure acronyms are expanded on first use, e.g. transcriptional repression-induced atypical cell death of neuron (TRIAD).

>>> We described full names of such acronyms including TRIAD, Ku70, HMGB1, ERK, MEK, PKC, ATR, ATM and so on, where they are firstly used in the text.

3. In the introduction: “DNA damage accumulates also in senescent cells, which secrete a set of chemokines and cytokines collectively known as senescence-associated secretory phenotype (SASP)^{21–24}”

a. The authors may consider citing a recent review on the SASP here for simplicity, ideally one that addresses SASP in stress-induced (in addition to oncogenic) senescence.

>>> We added one sentence here (in Introduction) and referred review papers related to the roles of SASP in stress-induced senescence.

4. Given the mention of senescence and the focus of the paper, a brief sentence on the presence of senescent glial and neuronal cells in neurodegeneration and the ageing brain should be included. See works from labs of Miranda Orr, Diana Jurk, Julie Andersen, etc. I have some more comments on this in points 8-10.

>>> We added another sentence here (in Introduction) and referred papers related to the presence of senescent glial and neuronal cells in neurodegeneration and the ageing brain.

5. Figure 6 B/C seem to be mixed up in the text. I would suggest double checking if that's correct.

>>> We separated the reference of Figure 6B and that of Figure 6C in the text. Also changed alignment of their panels in Figure 6.

6. Some of the text in the figures is quite small due to the amount of data and may be difficult for readers with visual impairments. The current resolution is sufficient that you can zoom in on them, however.

>>> We enlarged font size in figures as possible as we could.

7. Methods for Y-Maze should really state dimensions of the maze, phase of light-cycle the experiment was performed in (and preferably light intensity). For those from a behavioural background looking at

the data, handling technique (tail vs tube handling) and whether mice were allowed to acclimatise to the experimental room are also useful to know.

>>> We described dimensions of Y-maze and information about acclimatisation in Methods.

a. If it was performed during the light cycle, stress from handling, high light intensities and waking them during their sleep cycle can lead to increased freezing, defecation and time spent in the 'home arm' (i.e. the arm into which the mouse was initially placed). This can influence the assessment of working memory, and can drive alteration ratios below 50% (due to shuttling from the 'home arm' to one other arm, without exploration of the third arm). I would suggest the authors consider performing trialling the use of the Y-Maze during the dark phase if they have not already tried it (under dim or red-light, <2 lux), as they should get cleaner data on memory. In my own hands, control mice of this age (C57Bl/6) perform much better in this set up (80-100% alternation) in a 40cm long arm maze and 8 minute trial, while impaired mice still show poor performance.

>>> We appreciate kind advices on behavioral test from the reviewer. Yes, we performed Y-maze test at dark-cycle (20:00 - 8:00). But the darkness less than 20 lux used for our study may not be enough considering with the standard suggested by the reviewer. As the reviewer mention in the following, our condition is rather common and not exceptional in the field.

b. I would not consider this a barrier to publication (rather intended as a helpful suggestion for future work), as performing Y-maze during the light cycle is still common practice in the field. The data they show still looks convincing, but they could mention that there

may some confounding influences from anxiety. It would also be entirely unreasonable to ask them to repeat this part of the study

>>> We thank the reviewer very much for these kind advices with rich experiences of the reviewer. We performed Y-maze test at dark-cycle (20:00 - 8:00) in the darkness less than 20 lux. To trace the movement of mice, we have pre-checked the threshold of darkness in which the machine could chase the mouse movement. We added description about these conditions in the "Method".

We completely agree with the reviewer that our data will be improved more to exclude the concerns of mouse anxiety if it had been performed in the darkness less than 2 lux.

Though we believe that our data in our darkness condition are still valuable, we will surely take the advice and will improve our future experiments.

c. A dotted line through the 50% mark on the y-axis would also be preferred, to show expected result from random alternation (i.e. from dysfunctional memory).

>>> We added a dot line at 50%.

8. While the authors mention cellular senescence, the senescence associated secretory phenotype (SASP) and its role in the release of HMGB1 to the extracellular space in the introduction, they do not address it later in the paper. HMGB1 has been identified as important component of some senescent and SASP phenotypes, and this work may show how this could drive bystander DNA damage from senescent cells in some populations (previously largely

assumed to be due to reactive species production by senescent cells). Especially given the increasing evidence of the role of senescence in glial and neuronal populations driving early pathology in AD and cognitive decline, the authors could discuss this in the discussion.

>>> We thank very much for this kind evaluation of the significance of our paper showing that HMGB1 released from disease-associated or senescent cells further drives bystander DNA damage in surrounding cells.

Following the advice of the reviewer, we performed the first line of experiments for the effect of HMGB1 on senescence especially with regards to TRIAD necrosis (new Figure 7, 8).

9. It would be nice to have some evidence to address this in the paper. A relatively limited evaluation of senescent and inflammatory markers, and discussion would suffice if the authors wish to pursue this - any more in-depth investigation would be enough work to warrant another paper. It would be interesting to see if treatment with anti-HGMB1 antibody or Gö6976 alters levels of senescence markers (as might be expected, with interference with paracrine signalling and reduced DNA damage). If they choose this, I suggest sticking to a few relatively simple markers:

- p16 and p21 - can be determined in homogenate, or by RNA fluorescence in situ.
- Lamin B1 - can be performed in immunohistochemistry (positive/negative or intensity)
- Nuclear size - can be taken from existing images.
- γ H2A.X - data is already present.
- I would not suggest Sen- β -Gal in neuronal cells. HMGB1 sub-cellular localisation by immunohistochemistry (nuclear

exclusion) is also sometimes used, but as this paper shows it has many roles.

>>> We employed Lamin B1 and nuclear size from suggested senescence markers, and examined whether the treatment of anti-HGMB1 antibody or Gö6976 alters levels of such senescence markers (new Figure 7, 8). As expected by the reviewer, we also observed that anti-HGMB1 antibody inhibited senescence phenotypes of cells in the brain.

10. Assessment of inflammatory markers (senescence associated or otherwise) would also be nice to have, as reductions through interruption of this cascade and inflammatory and SASP-signalling may contribute towards the improved phenotype. Assessment of glial cells would also be nice, as HMGB1 released by neurons (amongst other cells) can drive chronic inflammation and neurodegeneration. Markers of microglia (e.g. iba1) and microglial activation could be considered, as well as astrocytes if time permits. Reductions in inflammation from reduced extracellular HMGB1 signalling may be partially behind the improvements seen in treated mice (I suspect it will actually be an interplay between all these factors in forming/breaking a vicious cycle of destruction).

>>> We also used inflammatory markers associated with senescence (MCP1 and IL-6), and examined whether the treatment of anti-HGMB1 antibody improves such senescence-associated inflammation (new Figure 8).

11. I would be interested in the author's speculation on why this mechanism exists, whether it has a physiological role or is purely aberrant (i.e. why have a DAMP signal that reduces DNA repair in recipient cells, perhaps useful in countering parasitic and viral

infection?). This may be a bit speculative for the discussion in this paper, I am simply interested in their thoughts.

>>> We appreciate very much the very stimulating idea suggested by the reviewer. I personally agree with the reviewer's hypothesis that DAMPs signals play a role in protection against infectious organisms and that this system malfunctions in endogenous parasite/virus-mimic accumulation of disease proteins within neurons and glia. This might be too speculative, but I would add some short sentences suggesting this idea.

Reviewer #2 (Remarks to the Author):

In this manuscript, Tanaka et al studied the mechanism of the increase in DNA damage in Alzheimer's Disease (AD). They used comprehensive phosphoproteome analysis and other techniques to study abnormal phosphorylation of Ku70 at Ser77/78 and its effects in Ku70-DNA interaction using human AD postmortem brains. While I am not familiar of this field and the novelty of this particular study with respect to the existing literature, it is evident to me that this a carefully executed research. The experiments are planned well and the results are nicely explained. The following are some of my concerns, which I request the authors to consider:

>>> Thank you very much for your kind and high evaluation of our paper.

a) One of the important inference of this paper has come from the simulations using Pymol. Pymol offers only limited scope for any

serious simulations, which needs to be complimented with a MD (or similar) simulations using a reasonable time scale.

>>> We performed MD simulation following the advice, and show the results in Figure 1c.

b) Some of the figures, which contain protein structures (for example, Extended data 1), are difficult to follow. Since these figures are important for this manuscript, the authors should find a way to increase the size and/or the resolutions.

>>> In new Supplementary Figure 2 (= old Extended data 1B), we added an inlay for phospho-Thr401. In addition, we have also increased the size of images by making an independent Supplementary Figure 2.

c) SPR experiments are not carefully analyzed. For example, there is a number (1.8×10^{-8} M) provided for the data of #213-012 (Extended data 2D), I am not sure from the figure, if these numbers are reliable. The authors should provide some estimate of their fitting reliability, like residual distributions etc. The authors have fitted the linear inhibition data (Extended data 2H, the red curve for RAGE) with a non-linear equation, which is not correct. The authors should increase the dose and achieve saturation, or they should remove these numbers.

>>> We added the data of residual distributions in Supplementary Figure 3E, which support soundness of our analyses. We agree with the reviewer for that fitting of the inhibition data may not appropriate, and we thank the reviewer very much for pointing out this error. We replaced then with the improved data in which curve fitting was correctly performed (Supplementary Figure 3H).

d) How many times the SPR experiments were carried out? Not a single table has any error these measurements. Neither there is any error bars in the figures for these measurements.

>>> We performed the SPR experiments three times.

As the reviewer might know, the graphs of SPR show raw data of continuous change of resonance values, so it is difficult to put error bars on the raw data.

Therefore we presented Kd values as the mean +/- standard error of three times of experiments, above the graphs in new Supplementary Figure 2D (= old Extended data 2D).

e) Since the binding parameters are determined mostly from SPR, it would be nice if some of these data could be complemented using another assay.

>>> We added new data obtained by ELISA-based calculation of Kd. As described in the text, the value deduced from ELISA matched well with that generated from SPR.

** See the Nature Portfolio author and referees' website at www.nature.com/authors for information about policies, services and author benefits

Communications Biology is committed to improving transparency in authorship. As part of our efforts in this direction, we are now requesting that all authors identified as 'corresponding author' create and link their Open Researcher and Contributor Identifier (ORCID) with their account on the Manuscript Tracking System prior to acceptance. ORCID helps the scientific community achieve unambiguous attribution of all scholarly contributions. You can create and link your ORCID from the home page of the Manuscript Tracking System by clicking on 'Modify my Springer Nature account' and following the instructions in the link below. Please also inform all co-authors that they can add their ORCID to their accounts and that they must do so prior to acceptance.

For more information please visit

<http://www.springernature.com/orcid>

If you experience problems in linking your ORCID, please contact the Platform Support Helpdesk.

Our flexible approach during the COVID-19 pandemic

If you need more time at any stage of the peer-review process, please do let us know. While our systems will continue to remind you of the original timelines, we aim to be as flexible as possible during the current pandemic.

COMMSBIO - This email has been sent through the Springer Nature Tracking System NY-610A-NPG&MTS

Confidentiality Statement: □□ This e-mail is confidential and subject to copyright. Any unauthorised use or disclosure of its contents is

prohibited. If you have received this email in error please notify our Manuscript Tracking System Helpdesk team at

<http://platformsupport.nature.com> .

Details of the confidentiality and pre-publicity policy may be found here <http://www.nature.com/authors/policies/confidentiality.html>

Privacy Policy | Update Profile

REVIEWERS' COMMENTS:

Reviewer #2 (Remarks to the Author):

I would like to thank the authors for their hard work and revisions to the manuscript. I would strongly recommend it for publication.

I find their work on the presence of senescence in multiple cell types and the observation of bystander effects interesting and exciting. It adds to the importance of an already excellent paper.

The overlap of senescence of TRIAD necrosis and senescence appears to be good fit as an eventual cell-death outcome of senescent neuronal cells during AD. This provides evidence for some questions I have personally been thinking about for a while regarding the eventual fate of senescent cells in the brain.

The reduction of senescence (especially pronounced in neurons) and inflammatory markers with anti-HMGB1 treatment is exciting. It is to my knowledge the first shown reduction of senescent cells in the brain by directly targeting a released signalling molecule, HMGB1 (which appears key to the disease and symptomatic progression). Currently published papers have relied on potentially much more aggressive treatments such as senolytics (inducing cell death of senescent cells) or, very early treatment with more broad intervention via targeting upstream signalling pathways. That interruption of HMGB1 signalling achieved this after symptomatic onset is promising for its applicability in improving symptoms, and reducing progression, in humans.

Regarding the Y-maze. I thank them for their expanded methods. Their work is well performed, being in the dark-cycle and as dim as light as they could achieve with their set-up. This set-up will have reduced any effect of anxiety upon their data, and I am confident in their conclusions.

I agree with the hypothesis that parasite/virus mimicry by accumulation of disease proteins within neuronal and glial cells as a potential reason for the induction of DAMP signalling. This would also correspond with what is seen in other cell types and senescence, with damage factors leading to the (seemingly aberrant) triggering of anti-parasite/virus innate immune pathways.

Reviewer #3 (Remarks to the Author):

The authors have adequately addressed my concerns. This manuscript may be accepted for publication.